# On Symmetric Losses for Policy Optimization with Noisy Preferences

**Soichiro Nishimori**                                      *nishimori@ms.k.u-tokyo.ac.jp*
*The University of Tokyo, Japan*
*RIKEN AIP, Japan*

**Yu-Jie Zhang**                                            *yujiez7@cs.washington.edu*
*RIKEN AIP, Japan*
*University of Washington, US*

**Thanawat Lodkaew**                                        *lodkaew@ms.k.u-tokyo.ac.jp*
*The University of Tokyo, Japan*
*RIKEN AIP, Japan*

**Masashi Sugiyama**                                        *sugi@k.u-tokyo.ac.jp*
*RIKEN AIP, Japan*
*The University of Tokyo, Japan*

**Reviewed on OpenReview:** *https://openreview.net/forum?id=cBWGLmSeao*

## Abstract

Optimizing policies based on human preferences is key to aligning language models with human intent. This work focuses on reward modeling, a core component in reinforcement learning from human feedback (RLHF), and offline preference optimization, such as direct preference optimization. Conventional approaches typically assume accurate annotations. However, real-world preference data often contains noise due to human errors or biases, which can be asymmetric. We propose a principled framework for robust policy optimization under noisy preferences based on the view of reward modeling as a binary classification problem. Specifically, we demonstrate that asymmetric preference noise can be effectively treated as symmetric noise under this framework. This viewpoint allows us to leverage symmetric losses, well known for their robustness to label noise in classification, for reward modeling, which leads to our Symmetric Preference Optimization (SymPO) method, a novel offline preference optimization algorithm. Theoretically, we prove that symmetric losses enable successful policy improvement even with noisy labels, as the resulting reward is rank-preserving—a property we identify as sufficient for policy improvement. Empirical evaluations on a synthetic dataset and real-world language model alignment tasks demonstrate that SymPO achieves competitive or higher performance than existing robust methods in high-noise scenarios.

## 1 Introduction

Policy optimization with human preferences aims to train a policy that aligns with human desires, given pairs of actions $(a_1, a_2)$ and annotations indicating which action is preferred ($a_1 \succ a_2$ or $a_1 \prec a_2$) (Ouyang et al., 2022; Stiennon et al., 2020; Rafailov et al., 2024). This paradigm has become increasingly prominent in language model alignment, where the goal is to develop models that behave according to human values, preferences, and instructions.

A central component of this process is reward modeling, which involves learning an underlying reward function from preference data. Reward modeling is the foundation for two major approaches to policy optimization using preference data: Reinforcement Learning from Human Feedback (RLHF) (Ouyang et al., 2022) and

offline preference optimization (Tang et al., 2024). In RLHF, a reward model is first trained and then used to fine-tune the policy via on-policy RL methods. In contrast, offline preference optimization directly learns the policy from the collected preference data based on a reward modeling objective, as exemplified by Direct Preference Optimization (DPO) (Rafailov et al., 2024).

Most existing methods assume that preference labels are accurate. However, real-world preference data often suffer from noise due to annotation errors or systematic biases (Gao et al., 2024). This issue has garnered particular attention in offline preference optimization (Gao et al., 2024; Chowdhury et al., 2024; Liang et al., 2024; Wu et al., 2024; Fisch et al., 2024). Despite the existence of numerous methods for handling such errors and biases, these methods either require prior knowledge of the noise rate (Chowdhury et al., 2024) or entail additional hyperparameters (Liang et al., 2024; Wu et al., 2024) to be tuned. Furthermore, they assume *symmetric noise* (van Rooyen et al., 2015), where the preferences will flip with equal probability for $a_1 \succ a_2$ and $a_1 \prec a_2$. However, preference noise is often *asymmetric* in practice. For instance, an annotator may systematically favor responses from GPT-4 (Achiam et al., 2023) over those from GPT-3 (Brown et al., 2020) regardless of quality, introducing one-sided bias.

In this paper, we propose a general framework for learning reward models from noisy preferences, grounded in the perspective of binary classification. Viewing reward modeling as risk minimization in binary classification provides both methodological and theoretical benefits.

Methodologically, this classification viewpoint provides a principled way to handle asymmetric noise and enables the development of a robust objective function for reward modeling. Specifically, it makes an inherent structure in preference data explicit: swapping the positions of input pairs flips the label. Leveraging this, we show that asymmetric and symmetric noise are equivalent in reward modeling (Sec. 4.1). For robust reward modeling, we then employ *symmetric losses*, $\ell : \mathbb{R} \to \mathbb{R}$, that satisfy the symmetric condition: $\ell(z) + \ell(-z) = K$ for some constant $K$ as shown in Fig. 1 (Sec. 4.2). Symmetric losses are proven to be robust against symmetric label noise in binary classification settings (van Rooyen et al., 2015; Charoenphakdee et al., 2019), and we extend this result to show their robustness to general asymmetric noise in reward modeling. This gives rise to our method: **Sym**metric **P**reference **O**ptimization

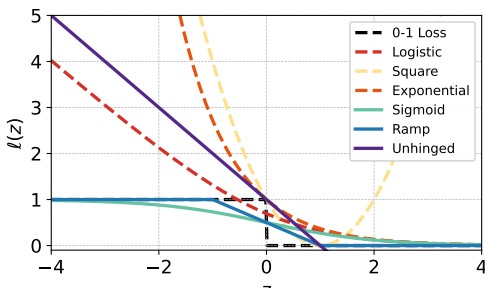

Figure 1: The symmetric losses are plotted with a solid line, while the $0-1$ loss and convex losses are plotted with a dashed line.

(**SymPO**), a novel offline preference optimization algorithm based on the symmetric loss.

In our theoretical analysis, we uncover a connection between a property of loss functions in binary classification and the success of policy optimization. We first pinpoint a sufficient condition of the reward for policy improvement in policy optimization: *rank preservation*, which ensures that the ordering of actions matches the true underlying reward function (Sec. 5.1). Next, we prove that minimizing the risk with *classification-calibrated losses* (Bartlett et al., 2006) leads to rank preservation, showing that a broad class of loss functions—including symmetric ones—is appropriate for policy optimization. By combining the robustness of symmetric losses with this policy improvement guarantee, we provide theoretical justification for our method, SymPO (Sec. 5.2).

Finally, we validate our approach through two sets of experiments: a synthetic setup based on MNIST and language model alignment tasks. In the MNIST setting, we evaluate the robustness of symmetric losses for reward modeling. Our language model alignment experiments show that SymPO facilitates robust policy improvement with noisy preference data.

**Contributions.** The main contributions of this paper are as follows: 1) We leverage the binary classification perspective of reward modeling to formally prove the equivalence between asymmetric and symmetric noise, and we propose a new offline preference optimization method, SymPO, that exploits the robustness of symmetric losses. 2) We bridge the gap between classification theory and policy optimization by proving that classification-calibrated losses yield policy improvement, thereby supporting the use of symmetric losses

in policy optimization. 3) We validate the robustness of symmetric losses in reward modeling and policy optimization through experiments.

## 2 Related Work

In this section, we review the related work in the field of reward modeling with noisy preference data and offline preference optimization.

### 2.1 Loss functions in Reward Modeling

Direct Preference Optimization (DPO) (Rafailov et al., 2024) aligns policies with human preferences by minimizing the cross-entropy or logistic loss under the assumption of the Bradley–Terry (BT) model (Bradley & Terry, 1952). Recently, many studies have explored alternatives and generalizations of DPO components (Tang et al., 2024; Azar et al., 2024; Zhao et al., 2023; Ethayarajh et al., 2024; Meng et al., 2024; Wu et al., 2025). For example, Azar et al. (2024) uses a squared loss to learn policies without relying on the BT model, while Zhao et al. (2023) employs a hinge loss to achieve more efficient preference optimization. Among these works, Generalized Preference Optimization (GPO) (Tang et al., 2024) is most closely related to our research. GPO frames preference optimization as a binary classification problem and examines various convex loss functions for their effectiveness in offline preference optimization. In contrast, our work focuses on non-convex loss functions, specifically exploring symmetric loss (van Rooyen et al., 2015). By doing so, we develop a robust reward learning method to label noise.

### 2.2 Reward Modeling with Noisy Preference Data

Several prior works focus on offline preference optimization with noisy preference labels (Gao et al., 2024; Fisch et al., 2024; Liang et al., 2024; Chowdhury et al., 2024; Ramesh et al., 2024; Yan et al., 2024; Bukharin et al., 2024; Choi et al., 2024; Fujisawa et al., 2025; Razin et al., 2025; Chen et al., 2025). These works can be broadly categorized according to their noise assumptions into two classes: instance-independent noise and instance-dependent noise. Under instance-independent noise, the probability that a preference label is corrupted is assumed to be uniform across instances, regardless of the prompt or response pair (Natarajan et al., 2013; Charoenphakdee et al., 2019). In contrast, under instance-dependent noise, the corruption probability varies across instances and may depend on properties of the prompt, the responses, or the preference pair itself (Cheng et al., 2020; Wu et al., 2022). In this work, we focus on the *instance-independent* setting as a principled starting point, because it makes the flipping structure of pairwise preferences analytically tractable and allows us to directly leverage the robustness properties of symmetric losses (van Rooyen et al., 2015).

For instance-independent noise, Chowdhury et al. (2024) studied preference optimization with noisy labels by adapting techniques from noisy label learning (Natarajan et al., 2013), while Wu et al. (2024) approached the problem from the perspective of distributionally robust optimization (DRO) (Duchi & Namkoong, 2019; 2021). The most closely related work is Robust Preference Optimization (ROPO) (Liang et al., 2024), which introduced a sigmoid-based noise-aware loss belonging to the class of symmetric losses. Their method interpolates between the original loss and the noise-aware loss, thereby avoiding the need for the noise-rate knowledge required by Chowdhury et al. (2024). Our work differs from ROPO in two main respects. First, we consider more general asymmetric noise and establish an equivalence between asymmetric and symmetric noise. Second, we provide a more comprehensive analysis of the learned reward function by leveraging existing binary classification theory, and we further establish a policy improvement guarantee for policy optimization with symmetric losses. Another line of work addresses noisy preferences through data filtering rather than loss design (Fujisawa et al., 2025; Liang et al., 2024). Since our primary focus is on the design and analysis of robust loss functions, we do not consider filtering-based approaches in this work. That said, as suggested by Liang et al. (2024), robust losses and filtering are not mutually exclusive, and combining the two is a promising direction for future research.

Instance-dependent noise has been studied in Bukharin et al. (2024); Razin et al. (2025); Chen et al. (2025). In particular, Bukharin et al. (2024) considered a specific form of instance-dependent noise and proposed an alternating procedure that estimates the noise rate while optimizing the policy, whereas Chen et al. (2025)

studied a likelihood-margin-based noise model. A data-filtering approach under instance-dependent noise has also been explored by Deng et al. (2025).

## 2.3 Weakly Supervised Learning

This work is inspired by weakly supervised learning in binary classification (Sugiyama et al., 2022). In this study, we address the reward modeling with preference noise given binary classification problems with noisy labels (Natarajan et al., 2013; Menon et al., 2015; van Rooyen et al., 2015). Natarajan et al. (2013) proposed a method for learning with noisy labels by utilizing noise rate information. van Rooyen et al. (2015) demonstrated the robustness of symmetric loss against symmetric noise. In deep learning, several studies have explored robust loss functions for handling label noise (Wang et al., 2019; Zhang & Sabuncu, 2018; Ghosh et al., 2017). Considering corrupted label learning as an extreme case of noisy labels, Charoenphakdee et al. (2019) and Menon et al. (2015) showed the robustness of symmetric loss in this setting. Regarding corrupted label learning, Lu et al. (2018) and Lu et al. (2020) proposed unlabeled-unlabeled (UU) learning, which constructs an unbiased estimator from two sets of unlabeled data with known class priors.

## 3 Preliminaries

Here, we provide a foundational knowledge for our work. Sec. 3.1 establishes reward modeling as the grounding concept for policy optimization from the human preferences. We explain the conventional way to learn reward function from human preferences based on the BT model, and demonstrate how it supports two major policy optimization paradigms: RLHF and offline preference optimization. Sec. 3.2 then casts reward modeling as the binary classification, preparing our approach to deal with noisy preferences.

### 3.1 Reward Modeling and Policy Optimization

In reward modeling with the BT model (Bradley & Terry, 1952), we assume for a pair of actions $(a_1, a_2) \in \mathcal{A} \times \mathcal{A}$ ($\mathcal{A} \subset \mathbb{R}^d$ and $|\mathcal{A}| < \infty$), the preference $a_1 \succ a_2$ is given by

$$p(a_1 \succ a_2) = \sigma(r_{\text{true}}(a_1) - r_{\text{true}}(a_2)), \tag{1}$$

where $r_{\text{true}} : \mathcal{A} \to \mathbb{R}$ is an underlying reward function and $\sigma$ is the sigmoid function.

**Reward Modeling.** Based on the BT model, given preference data in the form of $\mathcal{D} = \{(a_1^i \succ a_2^i)\}_{i=1}^n$, we train the reward model $r : \mathcal{A} \to \mathbb{R}$ by minimizing the following objective:

$$\mathcal{L}(r) = -\mathbb{E}_{(a_1, a_2) \sim \mathcal{D}} \left[ \log \sigma(r(a_1) - r(a_2)) \right], \tag{2}$$

where $\mathbb{E}_{(a_1, a_2) \sim \mathcal{D}} [\cdot]$ is the empirical average over the preference dataset $\mathcal{D}$. As explained subsequently, reward modeling plays a central role in two primary policy optimization approaches.

**Reinforcement Learning from Human Feedback (RLHF).** In RLHF (Ouyang et al., 2022), we first train a reward function via Eq. (2). Then, we optimize a policy $\pi \in \Delta(\mathcal{A})$ ($\Delta(\mathcal{A})$ is the probability simplex on $\mathcal{A}$) based on the Kullback-Leibler (KL) regularized reward maximization problem as [1]

$$\max_{\pi} \mathbb{E}_{a \sim \pi} \left[ r(a) \right] - \beta \text{KL}(\pi, \pi_{\text{ref}}), \tag{3}$$

where $\pi_{\text{ref}}$ is a reference policy, $\beta > 0$ is the temperature parameter controlling the strength of the regularization term, and $\text{KL}(p, q)$ is the KL divergence of $p$ from $q$. We optimize the policy using an on-policy RL algorithm such as PPO (Ouyang et al., 2022; Schulman et al., 2017).

---

[1]In conventional RLHF, the reward and policy takes some states, or prompts as the input, which we omit for brevity.

**Offline Preference Optimization.** Recently, DPO (Rafailov et al., 2024) was proposed to directly optimize the policy without the need for explicit reward modeling. They leveraged an analytical solution of the Eq. (3) to construct an *implicit reward* denoted as

$$r_{\text{imp}}^{\pi}(a) := \beta \log \frac{\pi(a)}{\pi_{\text{ref}}(a)}. \tag{4}$$

Then, by assigning it into the reward in Eq. (2), we have the policy optimization objective as

$$\mathcal{L}(\pi) := \mathcal{L}(r_{\text{imp}}^{\pi}) = -\mathbb{E}_{(a_1,a_2)\sim\mathcal{D}} \left[ \log \sigma \left( \beta \log \frac{\pi(a_1)}{\pi_{\text{ref}}(a_1)} - \beta \log \frac{\pi(a_2)}{\pi_{\text{ref}}(a_2)} \right) \right]. \tag{5}$$

Reward modeling lies at the heart of both RLHF and offline preference optimization, serving as a key mechanism in the former and an objective driver in the latter. Therefore, we center our study on reward modeling as the foundation of policy optimization from human preferences.

### 3.2 Reward Modeling as Binary Classification

Here, we show that the reward modeling problem can be cast as a binary classification task, enabling the use of diverse loss functions. For an input $(a_1, a_2) \in \mathcal{A} \times \mathcal{A}$, we define the label as $y = +1$ if $a_1 \succ a_2$, and $y = -1$ if $a_1 \prec a_2$ [2]. Let $p(a_1, a_2, y)$ be the joint density of input and label, $p(a_1, a_2)$ be the marginal density of input, and $p(y = +1) = \pi_{\text{p}}$ be the positive class prior. We emphasize a straightforward property of this labeling scheme: for any $(a_1, a_2)$ if we flip the position of $a_1$ and $a_2$, the label is flipped, e.g., if the label $y$ of $(a_1, a_2)$ is $+1 \iff a_1 \succ a_2$, then, the label of flipped input $(a_2, a_1)$ is $-1$. It plays a crucial role in handling the preference noise in Sec. 4.

Let $g : \mathcal{A} \times \mathcal{A} \to \mathbb{R}$ be the scoring function and $\ell : \mathbb{R} \to \mathbb{R}$ be the loss function. We optimize $g$ to minimize the $\ell$-risk defined as

$$R_\ell(g) := \mathbb{E}_{a_1,a_2,y}[\ell(y \cdot g(a_1, a_2))],$$

where $\mathbb{E}_{a_1,a_2,y}[\cdot]$ is the expectation over $p(a_1, a_2, y)$. In this study, we constrain $g(a_1, a_2)$ to have the form of $g(a_1, a_2) = r(a_1) - r(a_2)$ for some function $r : \mathcal{A} \to \mathbb{R}$. Then, the $\ell$-risk for reward modeling we consider can be expressed as

$$R_\ell(r) := \mathbb{E}_{a_1,a_2,y}[\ell(y(r(a_1) - r(a_2)))]. \tag{6}$$

We define the optimal $\ell$-risk as $R_\ell^* := \inf_r R_\ell(r)$, where infimum is taken over all measurable functions. As discussed in Tang et al. (2024), this formulation is natural generalization of the reward modeling objective defined in Eq. (2) because having $\ell(z) = \log(1 + e^{-z})$ (logistic loss) in Eq. (6) recovers the objective. Just as in Sec. 3.1, we can construct the offline preference optimization objective by putting $r_{\text{imp}}^{\pi}$ into $R_\ell(r)$.

## 4 Noisy Preference Label and Symmetric Loss

In this section, based on the binary classification framework in Sec. 3.2, we introduce an asymmetric noise model for preference labels and prove its equivalence to a symmetric noise model (Sec. 4.1). We then demonstrate the robustness of symmetric losses to the asymmetric noise in reward modeling and develop a corresponding preference optimization algorithm (Sec. 4.2).

### 4.1 Noise Model on Preference Label

We investigate the noisy preference label from the perspective of binary classification. Suppose we are given a dataset $\mathcal{D} = \{(a_1^i, a_2^i, \tilde{y}_i)\}_{i=1}^n$, where $\tilde{y}_i \in \{-1, +1\}$ is the noisy preference label. We assume the following

---

[2]We do not consider the case of tie $a_1 \sim a_2$ following the previous studies (Tang et al., 2024; Rafailov et al., 2024; Zhao et al., 2023; Azar et al., 2024)

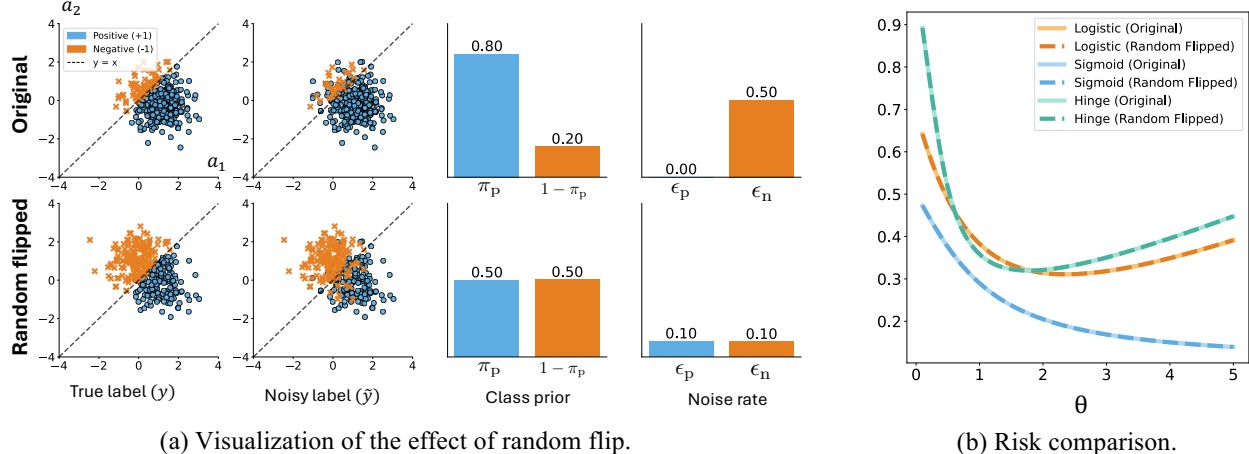

(a) Visualization of the effect of random flip.

(b) Risk comparison.

Figure 2: We sample action pairs $(a_1, a_2)$ from a 2D Gaussian and assign preference labels using $y = \text{sign}(a_1 - a_2)$. The class prior $\pi_{\text{p}}$ of the original data is 0.8. Asymmetric label noise is introduced with $\varepsilon_{\text{p}} = 0.0$ and $\varepsilon_{\text{n}} = 0.5$, producing noisy labels $\tilde{y}$. Figure (a) shows the action pairs before and after random flipping, along with class priors and noise rates. After noise is applied, the class prior becomes 0.5 and the overall error rate is $\pi_{\text{p}}\varepsilon_{\text{p}} + (1 - \pi_{\text{p}})\varepsilon_{\text{n}} = 0.1$. Figure (b) plots the empirical risk across different loss functions under noisy labels, revealing that the risk remains unchanged after random flipping. The loss is calculated as $\ell(y\theta(a_1 - a_2))$, with $\theta$ on the $x$-axis.

noise model:

$$p(\tilde{y} = -1|y = +1) = \varepsilon_{\text{p}}, \quad p(\tilde{y} = +1|y = -1) = \varepsilon_{\text{n}}, \tag{7}$$

where $\varepsilon_{\text{p}}, \varepsilon_{\text{n}} \in [0, 1.0)$ are the error rates. ROPO (Liang et al., 2024) and rDPO (Chowdhury et al., 2024) assume symmetric noise, where $\varepsilon_{\text{p}} = \varepsilon_{\text{n}}$ (van Rooyen et al., 2015). In contrast, our asymmetric noise model accounts for more realistic scenarios, such as systematically corrupted labels. When noisy labels are consistently flipped to either $+1$ or $-1$, the noise becomes fully asymmetric. Concretely, consider the case where $p(\tilde{y} = -1 \mid y = +1) = 0$, and $p(\tilde{y} = +1 \mid y = -1) = 1$, which can occur when an annotator consistently prefers outputs from a particular model (e.g., GPT-4 over GPT-3.5), irrespective of actual quality. However, in the following lemma, we can show that the risk with asymmetric noise is equivalent to that with symmetric noise by leveraging the flipping property of the preference label in Sec. 3.2.

**Lemma 1** (Equivalence of asymmetric and symmetric noise (Informal))**.** For any scoring function $g : \mathcal{A} \times \mathcal{A} \to \mathbb{R}$ such that $g(a_1, a_2) = -g(a_2, a_1)$, the risk for $g$ with the noise model in Eq. (7) is equivalent to that with symmetric noise for error rate $\pi_{\text{p}}\varepsilon_{\text{p}} + (1 - \pi_{\text{p}})\varepsilon_{\text{n}}$ with $p(y = +1) = 1/2$.

The intuition behind this lemma is that if the scoring function $g$ is symmetric, i.e., $g(a_1, a_2) = -g(a_2, a_1)$, then flipping the actions $a_1$ and $a_2$ does not change the value of the scoring function. If we consider randomly flipping all the actions with probability $1/2$ (we refer to this as *random flipping*), the class prior and error rates transform as described in the lemma. We illustrate this process in Fig. 2. The formal proof is provided in App. B.1. While prior work (Liang et al., 2024) also briefly mentioned this equivalence, we give a rigorous proof with the prerequisite conditions for the equivalence and the exact error rates of the symmetric noise model that is equivalent, in terms of the risk, to the original asymmetric noise model.

## 4.2 The Power of Symmetric Loss

Symmetric losses are functions that satisfy the property of $\ell(z) + \ell(-z) = K$, such as the sigmoid loss, unhinged loss, and ramp loss (Charoenphakdee et al., 2019). We define the $\ell$-risk with the noisy label $\tilde{y}$ similar to Eq. (6) as

$$\tilde{R}_\ell(r) := \mathbb{E}_{a_1, a_2, \tilde{y}}[\ell(\tilde{y}(r(a_1) - r(a_2)))], \tag{8}$$

where $\mathbb{E}_{a_1,a_2,\tilde{y}}[\cdot]$ is the expectation over the joint density over the input and noisy labels derived from $p(a_1, a_2, y)$ and the noise model defined in Eq. (7). The scoring function constructed from the reward, $r(a_1) - r(a_2)$, satisfies $g(a_1, a_2) = -g(a_2, a_1)$. Together with Lemma 1, we have the following proposition.

**Proposition 1** (Robust reward modeling under asymmetric noise). If the noisy labels are generated by the noise model in Eq. (7) and $\pi_{\mathrm{p}}\varepsilon_{\mathrm{p}} + (1 - \pi_{\mathrm{p}})\varepsilon_{\mathrm{n}} < 1/2$, for a symmetric loss $\ell_{\mathrm{sym}}$ and any reward functions $r, r'$, we have

$$\tilde{R}_{\ell_{\mathrm{sym}}}(r) > \tilde{R}_{\ell_{\mathrm{sym}}}(r') \iff R_{\ell_{\mathrm{sym}}}(r) > R_{\ell_{\mathrm{sym}}}(r'). \tag{9}$$

This proposition guarantees that the risk minimizer with the asymmetric noise model is equivalent to that with the clean labels in reward modeling. The proof of this proposition is based on van Rooyen et al. (2015) and the details are provided in App. B.1. Therefore, minimizing the risk with a symmetric loss provides a principled and theoretically justified solution for reward modeling under asymmetric noise. From a practical standpoint, this implies that learning is feasible even under highly asymmetric noise where either $\varepsilon_{\mathrm{p}}$ or $\varepsilon_{\mathrm{n}}$ exceeds 0.5. Furthermore, we can construct an offline policy optimization objective for noisy preference data as

$$\mathcal{L}_{\mathrm{sympo}}(\pi) := \mathbb{E}_{a_1,a_2,\tilde{y}}\left[\ell_{\mathrm{sym}}\left(\tilde{y}\left(\beta \log \frac{\pi(a_1)}{\pi_{\mathrm{ref}}(a_1)} - \beta \log \frac{\pi(a_2)}{\pi_{\mathrm{ref}}(a_2)}\right)\right)\right], \tag{10}$$

which we call **Sym**metric **P**reference **O**ptimization (SymPO).

## 5 Theoretical Analysis

In Sec. 4, we have shown the robustness of symmetric losses to the noisy preference label in reward modeling. In this section, we further guarantee the effectiveness of symmetric losses in policy optimization. Our ultimate goal is to fine-tune the policy $\pi$ to improve over the reference policy $\pi_{\mathrm{ref}}$ for some underlying true reward $r_{\mathrm{true}}$. We formalize this goal as follows:

**Definition 1** (Policy improvement). We say that the policy $\pi$ improves over the reference policy $\pi_{\mathrm{ref}}$ in terms of the true underlying reward $r_{\mathrm{true}}$ if

$$\mathbb{E}_{a\sim\pi}\left[r_{\mathrm{true}}(a)\right] > \mathbb{E}_{a\sim\pi_{\mathrm{ref}}}\left[r_{\mathrm{true}}(a)\right]. \tag{11}$$

Sec. 5.1 introduces the notion of a *rank-preserving reward* and shows that it is a sufficient condition for the policy improvement in both RLHF and offline preference optimization. Then, Sec. 5.2 shows the connection between the rank-preservingness and the classification-calibrated losses, which includes all common symmetric losses. This result provides theoretical support for using symmetric losses in policy optimization, thereby reinforcing the validity of our approach, SymPO. Furthermore, we explore the loss functions that can exactly recover the reward function in Sec 5.3.

**Assumptions.** We introduce the assumptions used in the theoretical analyses.

**Assumption 1** (Preference is consistent with the true reward). $2p(a_1 \succ a_2) > 1 \iff r_{\mathrm{true}}(a_1) > r_{\mathrm{true}}(a_2)$ for the true reward $r_{\mathrm{true}}$.

**Assumption 2** (Coverage of the probability distribution). For the marginal density $p(a_1, a_2)$ of the input space, we have $p(a_1, a_2) > 0$ for all $(a_1, a_2) \in \mathcal{A} \times \mathcal{A}$.

**Assumption 3** (Room for policy improvement). For at least two actions $a_1, a_2 \in \mathcal{A}$, $r_{\mathrm{true}}(a_1) \neq r_{\mathrm{true}}(a_2)$ and $\pi_{\mathrm{ref}}(a) > 0$ for all $a \in \mathcal{A}$.

The first assumption links the probability of a pairwise preference label to the true underlying reward. It is satisfied in the conventional preference model, e.g., the BT model in Eq. (1). The second assumption ensures to translate the property in expectation into that for all points in the input space. The third assumption guarantees the possibility of learning a better policy by assuming not all actions are equally good, and that the reference is not optimal, and covers the entire action space.

### 5.1 Rank-Preserving Reward and Policy Improvement

We introduce the notion of a *rank-preserving reward* as follows:

**Definition 2** (Rank-preserving reward). A reward $r$ is said to be *rank-preserving* w.r.t. another reward $r'$ if, for every pair of actions $(a_1, a_2) \in \mathcal{A} \times \mathcal{A}$, the ordering of their reward values is identical:

$$r'(a_1) > r'(a_2) \quad \implies \quad r(a_1) > r(a_2) \quad \forall (a_1, a_2) \in \mathcal{A} \times \mathcal{A}.$$

Now, we show that the rank-preservingness of the reward w.r.t. the true reward is a sufficient condition for policy improvement in two types of policy learning regimes.

We define the optimal policy of the RLHF problem in Eq. (3) w.r.t. a reward function $r$ and a reference policy $\pi_{\mathrm{ref}}$ as $\pi_r^* = \arg\max_\pi \mathbb{E}_{a \sim \pi}[r(a)] - \beta \mathrm{KL}(\pi, \pi_{\mathrm{ref}})$ for $\beta > 0$. The analytical form of the optimal policy is known (Rafailov et al., 2024). Therefore, the argmax is well-defined.

**Theorem 1** (Policy improvement in RLHF). With Assumption 3, for any reward that is rank-preserving w.r.t. $r_{\mathrm{true}}$, the policy improvement holds for the optimal policy over $\pi_{\mathrm{ref}}$ under $r_{\mathrm{true}}$.

The proof of this theorem is in App. B.2. This result implies that a reward function need not precisely match $r_{\mathrm{true}}$; it is sufficient to preserve the relative order of actions to achieve higher expected returns than the reference policy.

**Policy improvement in offline preference optimization.** In offline preference optimization, we have the relationship between our optimizing policy $\pi$ and the implicit reward as $r_{\mathrm{imp}}^\pi(a) = \beta \log \frac{\pi(a)}{\pi_{\mathrm{ref}}(a)}$. Therefore, it is straightforward to see that if the implicit reward $r_{\mathrm{imp}}^\pi$ is rank-preserving w.r.t. $r_{\mathrm{true}}$, the policy improvement holds, because it simply translates that the policy allocates the larger probability mass for the actions with higher reward compared with the reference policy. We provide the proof in App. B.3 for completeness.

### 5.2 Connection between Loss Function and Rank-Preserving Reward

In this subsection, we support the usage of symmetric losses in policy optimization by showing the connection between the rank-preservingness and the classification-calibrated losses, which include diverse loss functions such as the sigmoid, ramp, and unhinged losses.

**Classification-calibrated loss.** We invoke the concept of classification-calibrated loss (Bartlett et al., 2006) in binary classification and connect it to the rank-preserving reward. The classification calibration is known to be the minimal requirement for the loss function in binary classification (Bartlett et al., 2006). For the precise definition of the classification-calibrated loss, see App. B.4. To make the discussion clear, we assume that there exists the exact risk minimizer $r^*$, such that $R_\ell(r^*) = R_\ell^*$ [3]. Here, we show that if a loss $\ell$ is classification-calibrated, the minimizer of the $\ell$-risk is rank-preserving as follows:

**Theorem 2** (A classification-calibrated loss induces a rank-preserving reward). With Assumptions 1 and 2, if the loss function $\ell$ is classification-calibrated and the minimizer of the $\ell$-risk, $r^* = \arg\min_r R_\ell(r)$, exists, the minimizer is rank-preserving concerning the true reward.

Combined with the discussion in Sec. 5.1, we can see that the risk minimizer for the diverse classification-calibrated loss functions induces the policy improvement, making them effective for fine-tuning. This result bridges the minimal requirement for binary classification (classification calibration) with the fundamental goal in policy optimization (policy improvement), thereby reinforcing the solid connection between the two fields. Since all commonly used symmetric losses are known to be classification-calibrated (Charoenphakdee et al., 2019), and Proposition 1 confirms that the risk minimizer is invariant under noisy preferences, we conclude with a provable guarantee for *robust policy improvement by symmetric losses*, which is the primary focus of this theoretical analysis.

---

[3]If there is no exact minimizer, the same results hold for the sequences $r_1, r_2, \ldots$ that satisfy $R_\ell(r_i) \to R_\ell^*$ as $i \to \infty$ as shown in App. B.4.

Table 1: The properties of the loss functions.

| Loss | Function | Convex | Symmetric | CPE |
|---|---|---|---|---|
| Logistic | $\ell(z) = \log(1 + e^{-z})$ | Yes | No | Yes |
| Hinge | $\ell(z) = \max(0, 1 - z)$ | Yes | No | No |
| Squared | $\ell(z) = (z - 1)^2$ | Yes | No | Yes |
| Exponential | $\ell(z) = e^{-z}$ | Yes | No | Yes |
| **Sigmoid** | $\ell(z) = (1 + e^z)^{-1}$ | No | Yes | No |
| **Unhinged** | $\ell(z) = 1 - z$ | Yes | Yes | No |
| **Ramp** | $\ell(z) = \max(0, \min(1, 1/2 - 1/2z))$ | No | Yes | No |

**Corollary 1** (Robust policy improvement by symmetric losses)**.** With Assumptions 1, 2, and 3, if $\ell_{\text{sym}}$ is symmetric, and the noise is generated following the noise model in Eq (7), the policy improvement holds for

1. $\text{argmax}_\pi \mathbb{E}_{a \sim \pi} \left[ r^*(a) \right] - \beta \text{KL}(\pi, \pi_{\text{ref}})$, where $r^* = \text{argmin}_r \tilde{R}_{\ell_{\text{sym}}}(r)$ (RLHF),

2. $\text{argmin}_\pi \tilde{R}_{\ell_{\text{sym}}}(r_{\text{imp}}^\pi)$, where $r_{\text{imp}}^\pi(a) = \beta \log \frac{\pi(a)}{\pi_{\text{ref}}(a)}$ (offline preference optimization).

This corollary supports the application of a diverse set of binary classification loss functions to policy optimization. In conjunction with Proposition 1, it provides a theoretical guarantee for the policy improvement achieved by SymPO.

**Proof Sketch on Theorem 2.** We provide a proof sketch to explain how the classification-calibrated loss induces the rank-preserving reward. For the precise definition of the classification-calibrated loss and the complete proof of the theorem, see App. B.4. The classification-calibrated loss guarantees that the minimizer of the $\ell$-risk, $r^* = \text{argmin}_r R_\ell(r)$, is consistent with the Bayes classifier, $\text{sign}\left(2p(y = +1|a_1, a_2) - 1\right)$, as

$$\text{sign}\left(2p(y = +1|a_1, a_2) - 1\right) = \text{sign}\left(r^*(a_1) - r^*(a_2)\right).$$

Here we define $\text{sign}(0) := +1$, following the convention. In reward modeling, we have $p(y = +1|a_1, a_2) = p(a_1 \succ a_2)$ by the definition of labels and from Assumption 1, we have

$$\text{sign}\left(2p(y = +1|a_1, a_2) - 1\right) = \text{sign}\left(r_{\text{true}}(a_1) - r_{\text{true}}(a_2)\right),$$

which proves that the minimizer of the $\ell$-risk, $r^*$, is rank-preserving w.r.t. the true reward (QED).

### 5.3 Class Probability Estimation and Exact Reward Recovery

Although classification-calibrated losses guarantee the correct *ranking* of the risk minimizer concerning the true reward, one may still inquire whether the exact reward values can be recovered. To address this, we introduce the notion of *class probability estimation* (CPE) losses, whose minimizer recovers the precise posterior probability $p(y = +1|a_1, a_2) = p(a_1 \succ a_2)$ (Menon & Ong, 2016; Heagerty & Lele, 1998). As can be seen, if we know the relationship between the class probability and the reward, e.g., in the BT model, we can recover the exact reward. Some convex losses—such as the logistic and squared losses—are CPE losses, thus facilitating exact reward recovery. By contrast, *all symmetric losses are non-CPE*, as shown in Charoenphakdee et al. (2019). Consequently, while symmetric losses do not yield exact reward magnitudes, they preserve the action ranking. See App. B.5 for more details on CPE losses. For an overview of the properties of the losses, see Table 1.

## 6 Experiment

In this section, we demonstrate the effectiveness of symmetric losses for learning from noisy preference data. First, we verify the robustness of symmetric losses against noisy preference data in reward modeling using synthetic data. Then, using language model alignment tasks, we confirm robust policy improvement with symmetric losses in policy optimization to support the claims discussed in Sec. 5.

## 6.1 MNIST Preference Data

**Dataset.** MNIST Preference dataset is a synthetic preference dataset where the input is a pair of MNIST images, as shown in Fig. 3. The label is generated following the BT model, where the images' digits determine the rewards. We generated 10,000 pairs of images for training and 1,000 pairs for testing. To evaluate the effectiveness of the symmetric loss, we prepared the following symmetric noise: $\varepsilon \in \{0.0, 0.1, 0.2, 0.3, 0.4\}$. We injected the noise by randomly flipping the label with the given noise rate.

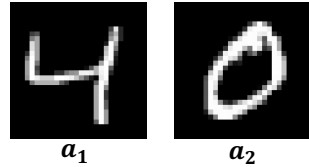

Figure 3: An instance of the MNIST Preference dataset.

**Training and Evaluation.** We compared sigmoid, ramp, and unhinged losses (symmetric) against logistic and hinge losses (non-symmetric) to evaluate the effectiveness of symmetric loss functions. A CNN classifier was trained using the Adam optimizer for one hundred epochs. For details such as hyperparameters, see App. A.1. Following reward learning conventions in language modeling, we report the test reward accuracy (correctly predicting the larger digit) averaged over five seeds.

**Results.** Table 2 presents the test reward accuracy of various losses under symmetric noise, with error rates $\varepsilon \in \{0.1, 0.2, 0.3, 0.4\}$. The symmetric losses, particularly ramp and sigmoid, are notably robust against high noise levels (e.g., $\varepsilon = 0.3, 0.4$) compared to the non-symmetric losses (logistic and hinge), suggesting the effectiveness of the symmetric loss functions in robust reward modeling.

Table 2: Average test reward accuracy (%).

| Loss | 0.0 | 0.1 | 0.2 | 0.3 | 0.4 |
|---|---|---|---|---|---|
| Logistic | 89.8 | 80.8 | 73.9 | 66.2 | 58.3 |
| Hinge | 89.4 | 81.2 | 74.5 | 66.5 | 57.6 |
| Unhinged | 77.7 | 76.9 | 74.0 | 70.0 | 61.6 |
| Ramp | 92.2 | 89.9 | 86.8 | 82.2 | 70.8 |
| Sigmoid | 91.2 | 88.9 | 85.6 | 80.9 | 68.8 |

## 6.2 Language Model Alignment

To assess the effectiveness of our method in more realistic settings, we performed language modeling experiments using standard benchmark datasets.

**Dataset and Training Setup.** We utilized two benchmark preference datasets: the Alpaca Comparison dataset (Peng et al., 2023) and the UltraFeedback Binarized (UFB) dataset (Cui et al., 2023). We used the full training set for both datasets (52,000 and 64,000 examples, respectively). Each example consists of a prompt paired with two responses: a chosen response and a rejected response. We employed the Llama-3.2 3B-Instruct model (Grattafiori et al., 2024) for all experiments. Following the procedure in Rafailov et al. (2024), we first trained a supervised fine-tuning (SFT) model on prompts and their corresponding chosen responses, then further fine-tuned the SFT model on prompts paired with both chosen and rejected responses. Each training stage was run for one epoch. To create noisy-label datasets, we varied the noise rate from 0.0 to 0.4 by randomly flipping the pairs of responses according to the specified ratios, following the previous work (Chowdhury et al., 2024; Liang et al., 2024). Note that as shown in Lemma 1, asymmetric preference noise is reducible to an equivalent symmetric noise model in preference learning. Therefore, symmetric corruption provides a controlled and sufficient testbed for validating the robustness predicted by our theory. For additional details, please refer to App. A.2. Training was performed on 8 NVIDIA A100 GPUs. For Alpaca, SFT and fine-tuning took approximately 20 and 36 minutes, respectively; for UFB, they took 28 and 49 minutes.

**Methods.** We selected two symmetric loss functions—sigmoid and ramp—as they achieved the highest accuracies in experiments with synthetic data. For the sigmoid and ramp losses, we clipped the output values within the range $[-20, 20]$ to ensure numerical stability for all the settings. Results with other clipping values (10 and 30) are provided in App. C.2, and we did not observe substantial differences. We compared these to three baselines: the standard DPO, which uses the logistic loss, and two noise-robust DPO variants—rDPO (Chowdhury et al., 2024) and ROPO (Liang et al., 2024). We set $\beta = 0.1$ for all methods. As rDPO constructs an unbiased estimator from noisy labels using the known error rate, we provided rDPO with the true error rate in our experiments. For ROPO, which includes additional components beyond the loss (i.e., rejection sampling and noisy sample filtering), we implemented only its loss function, a weighted combination of logistic

Table 3: Win rates (%) and standard deviations across 10 seeds on Alpaca Comparison (top) and Ultra-Feedback Binarized (UFB) (bottom) evaluated by GPT-4.1-nano and GPT-4o-mini. The noise rates are $\varepsilon \in \{0.0, 0.2, 0.4\}$. The best methods are highlighted in bold. Underlined scores indicate no statistical difference from the best method (5% t-test).

**Alpaca Comparison**

| Method | GPT-4.1-nano | | | GPT-4o-mini | | |
|---|---|---|---|---|---|---|
| | $\varepsilon = 0.0$ | 0.2 | 0.4 | 0.0 | 0.2 | 0.4 |
| DPO | $\underline{64.5}$ ±2.5 | $\underline{62.0}$ ±2.5 | 59.2 ±3.5 | $\underline{72.2}$ ±2.5 | 66.7 ±2.8 | 62.2 ±3.2 |
| rDPO | $\underline{64.1}$ ±2.5 | **64.3** ±2.5 | **64.1** ±2.8 | $\underline{72.1}$ ±2.3 | $\underline{70.3}$ ±1.7 | **69.5** ±2.3 |
| ROPO | **65.2** ±2.3 | $\underline{63.9}$ ±2.9 | $\underline{63.1}$ ±3.3 | **73.4** ±1.9 | $\underline{70.4}$ ±2.3 | $\underline{67.9}$ ±2.7 |
| Ramp (ours) | $\underline{64.9}$ ±2.1 | $\underline{64.0}$ ±2.2 | $\underline{63.1}$ ±3.0 | 70.8 ±2.5 | $\underline{69.8}$ ±1.8 | $\underline{67.7}$ ±2.5 |
| Sigmoid (ours) | $\underline{64.8}$ ±2.8 | **64.3** ±2.5 | $\underline{63.7}$ ±2.7 | $\underline{71.9}$ ±1.9 | **71.0** ±2.3 | $\underline{69.4}$ ±2.2 |
| SFT | 56.2 ±3.3 | | | 58.1 ±3.2 | | |

**UltraFeedback Binarized (UFB)**

| Method | GPT-4.1-nano | | | GPT-4o-mini | | |
|---|---|---|---|---|---|---|
| | $\varepsilon = 0.0$ | 0.2 | 0.4 | 0.0 | 0.2 | 0.4 |
| DPO | 56.2 ±1.2 | 54.4 ±0.9 | 52.4 ±1.2 | 60.5 ±1.3 | 58.5 ±1.1 | 55.8 ±1.2 |
| rDPO | 55.8 ±1.1 | 56.1 ±1.2 | 54.6 ±1.2 | 60.2 ±1.4 | 60.1 ±1.6 | 58.3 ±1.1 |
| ROPO | $\underline{57.7}$ ±1.2 | 57.7 ±0.7 | 55.1 ±1.4 | $\underline{63.2}$ ±1.1 | 62.3 ±1.3 | 59.2 ±1.0 |
| Ramp (ours) | $\underline{56.9}$ ±1.4 | 57.1 ±1.7 | $\underline{55.9}$ ±1.3 | 61.5 ±1.3 | 61.6 ±1.3 | 60.1 ±0.9 |
| Sigmoid (ours) | **58.8** ±1.1 | **59.1** ±1.5 | **56.8** ±1.2 | **63.6** ±1.5 | **64.4** ±1.4 | **61.3** ±1.0 |
| SFT | 51.3 ±1.0 | | | 54.7 ±1.5 | | |

and sigmoid losses, to enable controlled loss-level comparison under a unified training pipeline. Furthermore, to make the policy improvement, our main goal clearer, we also included the SFT model as a baseline. For further details on the baseline methods, please refer to App. A.2.

**Metrics.** We evaluated the effectiveness of all methods based on response generation quality, measured by comparing the helpfulness of the generated response to that of a reference response to train the SFT model following the procedure in Rafailov et al. (2024). Judgments were made using GPT-4o-mini and GPT-4.1-nano as evaluators. The prompt template used for evaluation is provided in App. A.2. After collecting judgments for all test examples, we calculated the win rate as the proportion of examples in which the generated response was preferred by the evaluator, relative to the total number of test examples. We generated the responses with 10 random seeds and conducted a t-test to verify the statistical significance across the performances of different methods.

**Results.** The results in Table 3 demonstrate that our methods are competitive with or superior to noise-robust baselines across different datasets. On the Alpaca dataset, we observed no statistically significant difference in win rates among the robust variants (rDPO, ROPO, and ours), indicating that our symmetric losses perform comparably to existing robust methods. However, on the UFB dataset, our method equipped with the sigmoid loss achieved the best performance. Notably, at high noise rates (e.g., $\varepsilon = 0.4$), it statistically significantly outperformed all other baselines, including rDPO, which was provided with the true noise rate. These results support our claim that the policy optimization based on symmetric losses achieves policy improvement even with noisy preference data. For further results, please refer to App. C.2.

# 7 Concluding Remarks

In this paper, we proposed a principled approach to learning reward functions from noisy preference data by framing reward modeling as a binary classification problem. We demonstrated the theoretical equivalence of asymmetric and symmetric preference noise, motivating the use of symmetric losses for improved robustness. Building on this, we introduced SymPO, a novel offline preference optimization algorithm. We provided a provable guarantee of policy improvement for SymPO by linking classification-calibrated losses to policy improvement. Empirical results further validated the robustness and effectiveness of symmetric losses on both synthetic datasets and language model alignment tasks. Regarding social impact, our proposed method is capable of improving policies even with noisy preference data, supported by provable guarantees. This enhances the applicability of offline preference optimization to real-world problems, where preference data is often noisy.

**Limitations and Future Work.** Our theoretical guarantee for policy improvement holds in the asymptotic regime and does not fully extend to the finite-sample case. Nevertheless, the established connection between classification-calibrated losses in binary classification and policy improvement in policy optimization offers valuable insight. Moreover, our analysis assumes instance-independent noise, whereas real-world preference data often exhibit instance-dependent noise. For example, annotations for similar responses are more likely to be incorrectly assigned than those for distinct responses. Therefore, extending our framework to accommodate instance-dependent noise (Cheng et al., 2020; Wu et al., 2022) is an important direction for future research. Furthermore, our instantiation of noise-robust policy optimization by symmetric losses is currently limited to offline preference alignment, and it would be worthwhile to explore the use of symmetric losses for reward modeling in on-policy RLHF methods (Schulman et al., 2017).

# 8 Acknowledgment

SN was supported by JSPS KAKENHI Grant Number JP24KJ0818. SN and MS were supported by JST ASPIRE Grant Number JPMJAP2405. TL was supported by the Institute for AI and Beyond at the University of Tokyo. We also thank Kazuki Ota for his insightful comments.

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

## A  Experimental settings

### A.1  MNIST Preference Dataset

We explain the experimental settings in this section.

**Datasets.**  We constructed the synthetic preference dataset using the MNIST (Deng, 2012) dataset. The MNIST dataset is available under the CC-BY-NC-SA 3.0 license. We randomly paired the images to form the preference dataset without pairs with the same digits. The reward is generated following the BT model with the reward being the digit of the image e.g. if the digit of the first image is 2 and the digit of the second image is 1, then the label is generated following $p(y = 1|\text{imgs}) = \sigma(2 - 1)$. We prepared 10000 pairs for training and 1000 pairs for testing. For introducing noise, we randomly flipped the label with probability $\varepsilon = 0.1, 0.2, 0.3, 0.4$.

**Hyperparameters.** For the reward model, we use a convolutional neural network (CNN) consisting of two convolutional layers followed by max pooling, a flattening layer, and three fully connected layers. We applied dropout with probability 0.2 and batch normalization for improved regularization and training stability. The final output is a single scalar, clipped to the range $[-20, 20]$. We trained the reward model for 100 epochs with a batch size of 512 and a learning rate of $1e - 3$.

## A.2 Language Model Alignment

This section explains the experimental settings for the language model alignment task.

**Datasets.** We used two publicly available datasets: Alpaca Comparison (Peng et al., 2023) and UltraFeedback Binarized (UFB) (Cui et al., 2023), both released under the MIT license, permitting unrestricted use. The Alpaca Comparison dataset comprises 52,000 instruction-following queries derived from the Stanford Alpaca dataset. For each query, candidate responses were generated using various models, including GPT-4 and text-davinci-003, which were subsequently scored by GPT-4 to establish preferences. The UltraFeedback Binarized dataset is a processed subset of the original UltraFeedback dataset. It comprises 64,000 prompts, each initially paired with four responses generated by large language models. Based on GPT-4 evaluations, two responses per prompt are selected to construct the binarized version for preference modeling tasks. From each dataset, we randomly sampled 20,000 pairs for training and 1,024 pairs for testing.

**Model.** Throughout the experiment, we used Llama-3.2 3B-Instruct (Grattafiori et al., 2024). The licence is provided in https://github.com/meta-llama/llama-models/blob/main/models/llama3_2/LICENSE.

**Baselines and Hyperparameters.** We list the loss functions for the baselines and our method. Given $a_1 \succ a_2$, we denote the logit $:= \log \frac{\pi(a_1)}{\pi_{\text{ref}}(a_1)} - \log \frac{\pi(a_2)}{\pi_{\text{ref}}(a_2)}$.

- DPO: $\ell_{\text{DPO}} = -\log \sigma(\beta \text{logit})$

- rDPO: $\ell_{\text{rDPO}} = -\frac{1-\varepsilon}{1-2\varepsilon} \log \sigma(\beta \text{logit}) - \frac{\varepsilon}{1-2\varepsilon} \log \sigma(-\beta \text{logit})$

- ROPO: $\ell_{\text{ROPO}} = \frac{4\alpha}{(1+\alpha)^2} \ell_{\text{DPO}} + \frac{4\alpha^2}{(1+\alpha)^2} \sigma(-\beta \text{logit})$

- Ramp (ours): $\ell_{\text{Ramp}} = \min(1, \max(0, \frac{1}{2} - \frac{\beta}{2} \text{logit}))$

- Sigmoid (ours): $\ell_{\text{Sigmoid}} = \sigma(-\beta \text{logit})$

We set $\beta = 0.1$ and the learning rate to be $5e - 6$ for all the methods. We trained the model for 1 epoch with a batch size of 32. For rDPO, we set $\varepsilon$ to be the true noise rate of the dataset. For ROPO, we set $\alpha$ to be 14 following the original paper Liang et al. (2024). As mentioned in the main body of the paper, we implemented only the loss function of ROPO, leaving aside the other parts of the algorithm (e.g., rejection sampling and noisy sample filtering).

**Implementation of SymPO.** For the reference implementation of SymPO, we provide the brief code block below.

```
def ramp_loss(
    pi_logp_chosen,
    pi_logp_rejected,
    ref_logp_chosen,
    ref_logp_rejected,
    beta=0.1
):
    logit = beta * (pi_logp_chosen - pi_logp_rejected) \
    - beta * (ref_logp_chosen - ref_logp_rejected)
```

```
        return torch.minimum(1, torch.maximum(0, 0.5 − logit))

def sigmoid_loss(
    pi_logp_chosen,
    pi_logp_rejected,
    ref_logp_chosen,
    ref_logp_rejected,
    beta=0.1
):
    logit = beta * (pi_logp_chosen − pi_logp_rejected) \
    − beta * (ref_logp_chosen − ref_logp_rejected)
    return torch.sigmoid(−logit)
```

**Evaluation.**   In the evaluation, we used the following prompt to compare the generated and reference responses.

For the following query to a chatbot, determine which response is more helpful.

\*\*Query:\*\* {query}

\*\*Response A:\*\* {response_A}

\*\*Response B:\*\* {response_B}

FIRST, provide a one−sentence comparison of the two responses, explaining which response is

SECOND, on a new line, state only "A" or "B" to indicate which response is more helpful.

Use the following format:

Comparison: <one−sentence comparison and explanation>

More helpful: <"A" or "B">

## B   Proofs

Here, we provide the proof of the lemmas, propositions, and theorems presented in the main body.

### B.1   Robustness to the Noisy Preferences

**Proof of Lemma 1.**   We have a scoring function $g : \mathcal{A} \times \mathcal{A} \to \mathbb{R}$ that satisfies $g(a_1, a_2) = -g(a_2, a_1)$. The label $Y$ is generated by some preference model $p(a_1 \succ a_2)$.

$$p(Y = +1|a_1, a_2) = p(a_1 \succ a_2), \quad \forall a_1, a_2 \in \mathcal{A},$$
$$p(Y = -1|a_1, a_2) = p(a_2 \succ a_1), \quad \forall a_1, a_2 \in \mathcal{A}.$$

Then, we define the noise model as

$$p(\tilde{Y} = -1|Y = +1) = \varepsilon_{\mathrm{p}}, \quad p(\tilde{Y} = +1|Y = -1) = \varepsilon_{\mathrm{n}},$$

where $\varepsilon_{\mathrm{p}}, \varepsilon_{\mathrm{n}} \in [0, 0.5)$.

To investigate the unique property of the preference label, we define the operation of flipping.

**Definition 3** (Flip Function). For a subset $\mathcal{S} \subseteq \mathcal{A} \times \mathcal{A}$, the flip function $f_{\mathcal{S}} : \mathcal{A} \times \mathcal{A} \times \{+1, -1\} \to \mathcal{A} \times \mathcal{A} \times \{+1, -1\}$ is defined as

$$f_{\mathcal{S}}(a_1, a_2, y) = \begin{cases} (a_2, a_1, -y) & \text{if } (a_1, a_2) \in \mathcal{S}, \\ (a_1, a_2, y) & \text{otherwise.} \end{cases}$$

We also define the flip function $h_{\mathcal{S}} : \mathcal{A} \times \mathcal{A} \times \{+1, -1\} \times \{+1, -1\} \to \mathcal{A} \times \mathcal{A} \times \{+1, -1\} \times \{+1, -1\}$ as

$$h_{\mathcal{S}}(a_1, a_2, y, y') = \begin{cases} (a_2, a_1, -y, -y') & \text{if } (a_1, a_2) \in \mathcal{S}, \\ (a_1, a_2, y, y') & \text{otherwise.} \end{cases}$$

We establish the equivalence of the risk under the flipping operation as follows.

**Lemma 2** (Flip Equivalence of Risk). For any scoring function $g : \mathcal{A} \times \mathcal{A} \to \mathbb{R}$ that satisfies $g(a_1, a_2) = -g(a_2, a_1)$, and a subset $\mathcal{S} \subseteq \mathcal{A} \times \mathcal{A}$, we have

$$\mathbb{E}_{p(a_1, a_2, y)} \left[ \ell(y \cdot g(a_1, a_2)) \right] = \mathbb{E}_{(a_1', a_2', y') = f_{\mathcal{S}}(a_1, a_2, y)} \left[ \ell(y' \cdot g(a_1', a_2')) \right].$$

*Proof.*

$$\begin{aligned} \mathbb{E}_{(a_1', a_2', y') = f_{\mathcal{S}}(a_1, a_2, y)} \left[ \ell(y' \cdot g(a_1', a_2')) \right] &= p(\mathcal{S}) \mathbb{E} \left[ \ell(-y \cdot g(a_2, a_1)) | \mathcal{S} \right] + p(\mathcal{S}^c) \mathbb{E} \left[ \ell(y \cdot g(a_1, a_2)) | \mathcal{S}^c \right] \\ &= p(\mathcal{S}) \mathbb{E} \left[ \ell(y \cdot g(a_1, a_2)) | \mathcal{S} \right] + p(\mathcal{S}^c) \mathbb{E} \left[ \ell(y \cdot g(a_1, a_2)) | \mathcal{S}^c \right] \\ &= \mathbb{E}_{p(a_1, a_2, y)} \left[ \ell(y \cdot g(a_1, a_2)) \right], \end{aligned}$$

where the second equality is due to $g(a_1, a_2) = -g(a_2, a_1)$. $\square$

Now, we show a conceptual flipping operation to reduce the asymmetric noise into a symmetric one.

**Lemma 3** (Flip symmetrization). For all $(a_1, a_2) \in \mathcal{A} \times \mathcal{A}$, we generate Bernoulli random variables $b_1, \ldots, b_{|\mathcal{A}| \times |\mathcal{A}|} \sim \text{Bernoulli} 1/2$. Let $\text{ind}(a_1, a_2)$ be the index function that maps $(a_1, a_2)$ to the index from $1$ to $|\mathcal{A}| \times |\mathcal{A}|$. We consider $\mathcal{P} = \{(a_1, a_2) \in \mathcal{A} \times \mathcal{A} : \text{ s.t. } b_{\text{ind}(a_1, a_2)} = 1\}$, and $h_{\mathcal{P}}$. Consider the random variable $(A_1', A_2', \tilde{Y}', Y') = h_{\mathcal{P}}(A_1, A_2, \tilde{Y}, Y)$, where $(A_1, A_2, \tilde{Y}, Y)$ follows the distribution defined above. Then, we have

$$P(Y' = +1) = \frac{1}{2},$$

$$P(\tilde{Y}' = -1 | Y' = +1) = P(\tilde{Y}' = +1 | Y' = -1) = \pi_{\text{p}} \varepsilon_{\text{p}} + (1 - \pi_{\text{p}}) \varepsilon_{\text{n}}.$$

*Proof.* We first show the first equation.

$$\begin{aligned} P(Y' = +1) &= p(\mathcal{P}) \mathbb{E} \left[ \mathbb{I}(Y = -1) | \mathcal{P} \right] + p(\mathcal{P}^c) \mathbb{E} \left[ \mathbb{I}(Y = +1) | \mathcal{P}^c \right] \\ &= \frac{1}{2} \pi_{\text{p}} + \frac{1}{2} (1 - \pi_{\text{p}}) \\ &= \frac{1}{2}. \end{aligned}$$

We then show the second equation.

$$\begin{aligned} P(\tilde{Y}' = -1 | Y' = +1) &= p(\mathcal{P}) \mathbb{E} \left[ \mathbb{I}(\tilde{Y} = +1) | \mathcal{P}^c, Y = -1 \right] + p(\mathcal{P}^c) \mathbb{E} \left[ \mathbb{I}(\tilde{Y} = -1) | \mathcal{P}^c, Y = +1 \right] \\ &= \frac{\frac{1}{2} \pi_{\text{p}} \varepsilon_{\text{p}} + \frac{1}{2} (1 - \pi_{\text{p}}) \varepsilon_{\text{n}}}{\frac{1}{2}} \\ &= \pi_{\text{p}} \varepsilon_{\text{p}} + (1 - \pi_{\text{p}}) \varepsilon_{\text{n}}. \end{aligned}$$

$\square$

Since the flip symmetrization can be achieved by the flip function, we have that the risk with the asymmetric noise model is equivalent to applying the flip symmetrization to the label, which is the risk with the symmetric noise model.

**Proof of Proposition 1.** From Lemma 1, for the scoring function $r(a_1) - r(a_2)$, the risk with asymmetric noise defined in Eq (7) is equivalent to that with a symmetric noise with error rate $\pi_{\mathrm{p}}\varepsilon_{\mathrm{p}} + (1 - \pi_{\mathrm{p}})\varepsilon_{\mathrm{n}}$.

Therefore, if $\pi_{\mathrm{p}}\varepsilon_{\mathrm{p}} + (1 - \pi_{\mathrm{p}})\varepsilon_{\mathrm{n}} < 1/2$, we can assume the noise model is symmetric as

$$P(\tilde{y} = +1|y = -1) = P(\tilde{y} = -1|y = +1) = \varepsilon, \quad \varepsilon \in [0, 0.5), \tag{12}$$

We introduce the noise-correct loss for a loss $\ell$, developed in Natarajan et al. (2013) as

$$\tilde{\ell}(z) = \frac{(1 - \varepsilon)\ell(z) - \varepsilon\ell(-z)}{1 - 2\varepsilon}.$$

From Lemma 1 of Natarajan et al. (2013), for the risk for clean label (Eq. (6)), with the noisy label (Eq. (8)), we have

$$\tilde{R}_{\tilde{\ell}}(r) = R_\ell(r). \tag{13}$$

The proof is based on Theorem 3 of van Rooyen et al. (2015), which guarantees if the $\ell$ is symmetric, for any $\varepsilon \in [0, 0.5)$, data distribution $p(a_1, a_2, y)$, and reward functions $r, r'$, we have

$$\underbrace{\mathbb{E}_{p(a_1,a_2,y)}\left[\tilde{\ell}(y(r(a_1) - r(a_2)))\right] > \mathbb{E}_{p(a_1,a_2,y)}\left[\tilde{\ell}(y(r'(a_1) - r'(a_2)))\right]}_{(A)},$$

if and only if

$$\underbrace{\mathbb{E}_{p(a_1,a_2,y)}\left[\ell(y(r(a_1) - r(a_2)))\right] > \mathbb{E}_{p(a_1,a_2,y)}\left[\ell(y(r'(a_1) - r'(a_2)))\right]}_{(B)}.$$

So, taking $p(a_1, a_2, y) = p(a_1, a_2, \tilde{y})$ with noise model as in Eq. (12), for (B), we have

$$\tilde{R}_\ell(r) > \tilde{R}_\ell(r').$$

Then, From Eq. (13), as for (A), we have

$$\mathbb{E}_{p(a_1,a_2,\tilde{y})}\left[\tilde{\ell}(\tilde{y}(r(a_1) - r(a_2)))\right] = \mathbb{E}_{p(a_1,a_2,y)}\left[\ell(y(r(a_1) - r(a_2)))\right]$$
$$= R_\ell(r).$$

Therefore, we have

$$\tilde{R}_\ell(r) > \tilde{R}_\ell(r') \iff R_\ell(r) > R_\ell(r').$$

### B.2 Policy Improvement

**Proof of Theorem 1** We consider the regularized reward maximization problem for a reward $r$ and a reference policy $\pi_{\mathrm{ref}}$ defined in Eq. (3) as

$$\pi_r^* = \mathrm{argmax}_\pi \mathbb{E}_{a \sim \pi}\left[r(a)\right] - \beta \mathrm{KL}(\pi, \pi_{\mathrm{ref}}).$$

Our goal is to show that for any $r_{\mathrm{true}}$, such that satisfies Assumption 3: for at least two actions $a_i, a_j \in \mathcal{A}$ with $r_{\mathrm{true}}(a_i) \neq r_{\mathrm{true}}(a_j)$, and the reference policy $\pi_{\mathrm{ref}}(a) > 0$ for all $a \in \mathcal{A}$, if $r$ is rank-preserving with respect to $r_{\mathrm{true}}$, then, the optimal policy for $r$, $\pi_r^*$ achieves the policy improvement over $\pi_{\mathrm{ref}}$.

$$\sum_{a \in \mathcal{A}} r_{\mathrm{true}}(a)\pi_r^*(a) > \sum_{a \in \mathcal{A}} r_{\mathrm{true}}(a)\pi_{\mathrm{ref}}(a).$$

Since we have the finite action space, $\mathcal{A}, |\mathcal{A}| = K$, for simplicity, we assume that $r_{\text{true}}(a_1) \geq r_{\text{true}}(a_2) \geq \cdots \geq r_{\text{true}}(a_K)$ and for at least two actions, $r_{\text{true}}(a_i) > r_{\text{true}}(a_j)$ with $i < j$. We also assume that $r$ is rank-preserving with respect to $r_{\text{true}}$: for any $a, a'$, if $r_{\text{true}}(a) > r_{\text{true}}(a')$ then $r(a) > r(a')$. We proceed with the proof based on the vector form of the reward and policy.

We first use the analytical form of the optimal policy $\pi_r^*$ as:

$$\pi_r^*(a_k) = \frac{\exp\left(\frac{r(a_k)}{\beta}\right) \pi_{\text{ref}}(a_k)}{Z},$$

where $Z$ is the normalization constant.

Now, we define the groups of actions with different true reward values $v_1 > v_2 \ldots, > v_M, M \leq K$ as:

$$\mathcal{A}_i := \{a \in \mathcal{A} \mid r_{\text{true}}(a) = v_i\}.$$

From the assumption that for at least two actions, $r_{\text{true}}(a_i) > r_{\text{true}}(a_j)$, we have $M \geq 2$.

Now, we rewrite the expected reward of the optimal policy and the reference policy as follows:

$$\sum_{a \in \mathcal{A}} r_{\text{true}}(a)\pi_r^*(a) = \sum_{i=1}^{M} v_i \sum_{a \in \mathcal{A}_i} \pi_r^*(a).$$

$$\sum_{a \in \mathcal{A}} r_{\text{true}}(a)\pi_{\text{ref}}(a) = \sum_{i=1}^{M} v_i \sum_{a \in \mathcal{A}_i} \pi_{\text{ref}}(a).$$

We analyze the difference between the expected reward of the optimal policy and the reference policy:

$$\sum_{a \in \mathcal{A}} r_{\text{true}}(a)\pi_r^*(a) - \sum_{a \in \mathcal{A}} r_{\text{true}}(a)\pi_{\text{ref}}(a) = \sum_{i=1}^{M} v_i \underbrace{\left(\sum_{a \in \mathcal{A}_i} \pi_r^*(a) - \sum_{a \in \mathcal{A}_i} \pi_{\text{ref}}(a)\right)}_{:=\Delta_i}.$$

From the fact that $\pi_{\text{ref}}$ and $\pi_r^*$ are both distributions, we have $\sum_{i=1}^{M} \Delta_i = 0$.

Now, we separate $\{1, \ldots, M\}$ into three parts:

$$\mathcal{I}_+ := \{i \mid \Delta_i > 0\}, \mathcal{I}_0 := \{i \mid \Delta_i = 0\}, \mathcal{I}_- := \{i \mid \Delta_i < 0\},$$

where $\mathcal{I}_+$ and $\mathcal{I}_-$ are non-empty by Lemma 5.

Then, we have:

$$\sum_{a \in \mathcal{A}} r_{\text{true}}(a)\pi_r^*(a) - \sum_{a \in \mathcal{A}} r_{\text{true}}(a)\pi_{\text{ref}}(a) = \sum_{i=1}^{M} v_i \Delta_i$$

$$= \sum_{j \in \mathcal{I}_+} v_j \Delta_j + \sum_{k \in \mathcal{I}_0} v_k \Delta_k + \sum_{l \in \mathcal{I}_-} v_l \Delta_l$$

$$= \sum_{j \in \mathcal{I}_+} v_j \Delta_j + \sum_{l \in \mathcal{I}_-} v_l \Delta_l.$$

Define $v_{\min}^+ = \min_{i \in \mathcal{I}_+} v_i$ and $v_{\max}^- = \max_{j \in \mathcal{I}_-} v_j$. Then, we have from the Lemma 4 and the fact that $\sum_{i=1}^M \Delta_i = 0$ that:

$$\sum_{j \in \mathcal{I}_+} v_j \Delta_j + \sum_{l \in \mathcal{I}_-} v_l \Delta_l \geq v_{\min}^+ \sum_{j \in \mathcal{I}_+} \Delta_j + v_{\max}^- \sum_{l \in \mathcal{I}_-} \Delta_l > 0.$$

This completes the proof.

**Lemma 4.**

$$\min_{i \in \mathcal{I}_+} v_i > \max_{j \in \mathcal{I}_-} v_j.$$

*Proof.* We prove this lemma by contradiction. Suppose that $\min_{i \in \mathcal{I}_+} v_i \leq \max_{j \in \mathcal{I}_-} v_j$. Then, we have $\exists i \in \mathcal{I}_+$ and $j \in \mathcal{I}_-$ such that $v_i \leq v_j$. And from the construction of $v_1, \ldots, v_M$, we have $v_i < v_j$. Define $r_i^{\max} = \min_{a \in \mathcal{A}_i} r(a)$ and $r_j^{\min} = \min_{a \in \mathcal{A}_j} r(a)$. Then, from rank-preservingness of $r$ with respect to $r_{\text{true}}$, we have $r_i^{\max} < r_j^{\min}$. For $\Delta_i$, we have:

$$\begin{aligned}
\Delta_i &= \sum_{a \in \mathcal{A}_i} \pi_r^*(a) - \sum_{a \in \mathcal{A}_i} \pi_{\text{ref}}(a) \\
&= \sum_{a \in \mathcal{A}_i} \frac{\exp\left(\frac{r(a)}{\beta}\right) \pi_{\text{ref}}(a)}{Z} - \sum_{a \in \mathcal{A}_i} \pi_{\text{ref}}(a) \\
&\leq \sum_{a \in \mathcal{A}_i} \frac{\exp\left(\frac{r_i^{\max}}{\beta}\right) \pi_{\text{ref}}(a)}{Z} - \sum_{a \in \mathcal{A}_i} \pi_{\text{ref}}(a) \\
&= \left(\frac{\exp\left(\frac{r_i^{\max}}{\beta}\right)}{Z} - 1\right) \sum_{a \in \mathcal{A}_i} \pi_{\text{ref}}(a) \\
&< \left(\frac{\exp\left(\frac{r_j^{\min}}{\beta}\right)}{Z} - 1\right) \sum_{a \in \mathcal{A}_i} \pi_{\text{ref}}(a) \\
&\leq \Delta_j.
\end{aligned}$$

Since $i \in \mathcal{I}_+$, we have $j \in \mathcal{I}_+$ but it contradicts the assumption that $j \in \mathcal{I}_-$ and $v_i < v_j$. Therefore, we have $\min_{i \in \mathcal{I}_+} v_i > \max_{j \in \mathcal{I}_-} v_j$. $\qquad\square$

**Lemma 5.** Suppose we have the following conditions:

1. For at least two actions $a_i, a_j \in \mathcal{A}$ with $r_{\text{true}}(a_i) \neq r_{\text{true}}(a_j)$

2. $\pi_{\text{ref}}(a) > 0$ for all $a \in \mathcal{A}$.

Then, $\mathcal{I}_+$ and $\mathcal{I}_-$ are non-empty.

*Proof.* From the first condition, we have two actions $a_i, a_j \in \mathcal{A}$ with $r_{\text{true}}(a_i) \neq r_{\text{true}}(a_j)$. Let assume $r_{\text{true}}(a_i) > r_{\text{true}}(a_j)$, then from the rank-preservingness of $r$ with respect to $r_{\text{true}}$, we have $r(a_i) > r(a_j)$. Also, from the second condition, we have $\pi_{\text{ref}}(a_i) > 0$. Then, from the normalization condition, we have

$$\pi_r^*(a_i) - \pi_{\text{ref}}(a_i) = \frac{\exp\left(\frac{r(a_i)}{\beta}\right) \pi_{\text{ref}}(a_i)}{Z} - \pi_{\text{ref}}(a_i) > 0$$

$$\pi_r^*(a_j) - \pi_{\text{ref}}(a_j) = \frac{\exp\left(\frac{r(a_j)}{\beta}\right) \pi_{\text{ref}}(a_j)}{Z} - \pi_{\text{ref}}(a_j) < 0.$$

Therefore, $\mathcal{I}_+$ and $\mathcal{I}_-$ are non-empty. $\qquad\square$

### B.3 Policy Improvement for Offline Preference Optimization

Here, we show that the rank-preservingness of the implicit reward $r_{\text{imp}}^{\pi}$ with respect to $r_{\text{true}}$ is equivalent to the policy improvement for offline preference optimization.

*Proof.* Let $A$ be the (finite) action set and write

$$w(a) := \frac{\pi(a)}{\pi_{\text{ref}}(a)}, \qquad a \in A.$$

**Step 1: Co-monotonicity.** The rank-preservingness of $r_{\text{imp}}^{\pi}$ with respect to $r_{\text{true}}$

$$r_{\text{true}}(a_1) > r_{\text{true}}(a_2) \implies r_{\text{imp}}^{\pi}(a_1) > r_{\text{imp}}^{\pi}(a_2), \quad \forall a_1 \in \mathcal{A}, \forall a_2 \in \mathcal{A},$$

$$\implies \log\frac{\pi(a_1)}{\pi_{\text{ref}}(a_1)} > \log\frac{\pi(a_2)}{\pi_{\text{ref}}(a_2)}, \quad \forall a_1 \in \mathcal{A}, \forall a_2 \in \mathcal{A},$$

$$\implies \frac{\pi(a_1)}{\pi_{\text{ref}}(a_1)} > \frac{\pi(a_2)}{\pi_{\text{ref}}(a_2)}, \quad \forall a_1 \in \mathcal{A}, \forall a_2 \in \mathcal{A},$$

is equivalent to

$$r_{\text{true}}(a_1) > r_{\text{true}}(a_2) \implies w(a_1) > w(a_2), \quad \forall a_1 \in \mathcal{A}, \forall a_2 \in \mathcal{A}.$$

**Step 2: Relating expectations to a covariance.** Because $\sum_{a \in A} \pi_{\text{ref}}(a)w(a) = 1$, the random variable $W := w(A)$ with $A \sim \pi_{\text{ref}}$ satisfies $\mathbb{E}_{\pi_{\text{ref}}}[W] = 1$. Define $R := r_{\text{true}}(A)$. Then

$$\mathbb{E}_{\pi}[r_{\text{true}}(A)] = \sum_{a \in A} r_{\text{true}}(a)\, \pi(a) = \sum_{a \in A} r_{\text{true}}(a)\, w(a)\, \pi_{\text{ref}}(a) = \mathbb{E}_{\pi_{\text{ref}}}[RW].$$

Subtracting $\mathbb{E}_{\pi_{\text{ref}}}[R]$ on both sides gives the identity

$$\mathbb{E}_{\pi}[r_{\text{true}}(A)] - \mathbb{E}_{\pi_{\text{ref}}}[r_{\text{true}}(A)] = \underbrace{\mathbb{E}_{\pi_{\text{ref}}}[RW] - \mathbb{E}_{\pi_{\text{ref}}}[R]\,\mathbb{E}_{\pi_{\text{ref}}}[W]}_{= \text{Cov}_{\pi_{\text{ref}}}(R,W)}.$$

**Step 3: Covariance is non-negative.** For any distribution $\mu$, co-monotone random variables have non-negative covariance. Since $r_{\text{true}}$ and $w$ are strictly co-monotone,

$$\text{Cov}_{\pi_{\text{ref}}}(R, W) \geq 0,$$

and the inequality is strict whenever $r_{\text{true}}$ is not constant on $A$.

Inserting the covariance bound into (B.3) yields

$$\mathbb{E}_{\pi}[r_{\text{true}}(A)] \geq \mathbb{E}_{\pi_{\text{ref}}}[r_{\text{true}}(A)],$$

with strict inequality whenever $r_{\text{true}}$ is non-constant, which is guaranteed by the Assumption 3. □

**Remark on related theoretical arguments.** Prior literature has primarily investigated *policy invariance*—identifying conditions under which reward transformations leave the optimal policy unchanged (Ng et al., 1999; Rafailov et al., 2024). Specifically, Ng et al. (1999) established necessary and sufficient conditions for potential-based reward shaping to maintain optimality in MDPs. Similarly, in the context of RLHF, Rafailov et al. (2024) showed that the optimal policy under a KL-constrained objective remains invariant to state-dependent reward shifts (i.e., $r'(x,y) = r(x,y) + f(x)$). In contrast, our work focuses on *policy*

*improvement.* We do not require the optimal policy to remain invariant; rather, we demonstrate that as long as the reward transformation is *rank-preserving* with respect to the true reward, the induced optimal policy $\pi_r^*$ strictly outperforms the reference policy $\pi_{\text{ref}}$. This distinction is crucial: unlike invariance results which concern the stability of the optimal solution, our theorem bridges the gap between classification metrics (ranking accuracy) and control performance, ensuring that rank-preserving rewards guarantee a strict performance gain.

### B.4 Rank Preservation and Reward Recoverability

Here, we provide proof of the rank preservation and reward recoverability.

**Proof of Theorem 2.** We first define the classification-calibrated loss to prove the theorem and show that it induces the rank-preserving reward. Below are some definitions necessary to define the classification-calibrated loss. Consider a binary classification setup with an input $x \in \mathcal{X}$ and a label $y \in \{-1, +1\}$. Let $\eta(x) := P(Y = +1 \mid x)$ be the instance-conditional posterior of the positive class. For $x \in \mathcal{X}$ taking $\eta = \eta(x)$ and $\alpha = g(x)$, we consider the conditional $\ell$-risk as

$$C_\eta^\ell(g) = \eta\ell(\alpha) + (1 - \eta)\ell(-\alpha),$$

and for $\eta \in [0, 1]$, we define the optimal conditional risk as

$$H_\eta^\ell(x) = \inf_{\alpha \in \mathbb{R}} C_\eta^\ell(\alpha).$$

We can define the classification-calibrated loss as

**Definition 4** (Classification-Calibrated Loss). A loss function $\ell$ is said to be classification-calibrated if for a sequence $\alpha_1, \alpha_2, \ldots$ that satisfies $\lim_{i \to \infty} \alpha_i = H_\eta^\ell(x)$, we have

$$\liminf_{i \to \infty} \alpha_i \text{sign}(2\eta - 1) = 1.$$

Therefore, if there is the exact minimizer $\alpha^*$ that satisfies $H_\eta^\ell(x) = C_\eta^\ell(\alpha^*)$, then if $\ell$ is classification-calibrated, we have

$$\text{sign}(\alpha^*) = \text{sign}(2\eta - 1).$$

For more details, see (Bartlett et al., 2006).

Now, we show that the classification-calibrated loss induces the rank-preserving reward in the sense of limit.

**Lemma 6** (Classification-calibrated loss induces rank-preserving reward). Under the Assumption 1 and 2, if $\ell$ is classification-calibrated and for the sequence of the reward $r_1, r_2, \ldots$, that satisfies $\lim_{i \to \infty} R_\ell(r_i) = R_\ell^*$, then we have

$$\lim_{i \to \infty} \text{sign}(r_i(a_1) - r_i(a_2)) = \text{sign}(r_{\text{true}}(a_1) - r_{\text{true}}(a_2)), \quad \forall a_1, a_2 \in \mathcal{A}.$$

Now, we can see that the Theorem 2 is the special case of the Lemma 6, where the exact minimizer of the $\ell$-risk can be achieved.

*Proof.* Define the zero-one risk as

$$R(r) = \mathbb{E}_{a_1, a_2, y}\left[\mathbb{I}(\text{sign}(r(a_1) - r(a_2)) = y)\right],$$

and its infimum as $R^* = \inf_r R(r)$, where the $\inf_r$ is taken over all the measurable functions. From Theorem 3 of (Bartlett et al., 2006), if $\ell$ is classification-calibrated, we have

$$\lim_{i \to \infty} R_\ell(r_i) = R_\ell^* \implies \lim_{i \to \infty} R(r_i) = R^*.$$

Since $R^*$ is achieved by the Bayes classifier,let $\eta(a_1, a_2) = P(Y = +1|a_1, a_2)$, we have

$$
\begin{aligned}
\lim_{i \to \infty} R(r_i) - R^* &= \lim_{i \to \infty} \mathbb{E}_{a_1, a_2, y} \left[ \mathbf{1}(\text{sign}(r_i(a_1) - r_i(a_2)) = y) - \mathbf{1}(\text{sign}(2\eta(a_1, a_2) - 1) = y) \right] \\
&= \lim_{i \to \infty} \mathbb{E}_{a_1, a_2, y} \left[ \mathbf{1}(\text{sign}(r_i(a_1) - r_i(a_2)) \neq \text{sign}(2\eta(a_1, a_2) - 1))|2\eta(a_1, a_2) - 1| \right] \\
&= 0,
\end{aligned}
$$

where $\mathbf{1}(\cdot)$ is the indicator function. From Assumption 1 that $\text{sign}(2\eta(a_1, a_2) - 1) = \text{sign}(r_{\text{true}}(a_1) - r_{\text{true}}(a_2))$, we have

$$
\lim_{i \to \infty} \mathbb{E}_{a_1, a_2, y} \left[ \mathbf{1}(\text{sign}(r_i(a_1) - r_i(a_2))) \neq \text{sign}(r_{\text{true}}(a_1) - r_{\text{true}}(a_2))|2\eta(a_1, a_2) - 1| \right] = 0.
$$

Since $|\mathcal{A}|$ is finite, we have

$$
\sum_{a_1, a_2 \in \mathcal{A}} p(a_1, a_2) \lim_{i \to \infty} \mathbf{1}(\text{sign}(r_i(a_1) - r(_i a_2))) \neq \text{sign}(r_{\text{true}}(a_1) - r_{\text{true}}(a_2))|2\eta(a_1, a_2) - 1| = 0.
$$

From Assumption 2 that $p(a_1, a_2) > 0$ for all $a_1, a_2 \in \mathcal{A}$, we have

$$
\lim_{i \to \infty} \mathbf{1}(\text{sign}(r_i(a_1) - r_i(a_2))) \neq \text{sign}(r_{\text{true}}(a_1) - r_{\text{true}}(a_2))|2\eta(a_1, a_2) - 1| = 0, \quad \forall a_1, a_2 \in \mathcal{A}.
$$

If $r_{\text{true}}(a_1) > r_{\text{true}}(a_2)$, from Assumption 1, we have $2\eta(a_1, a_2) - 1 > 0$ ,therefore,

$$
\lim_{i \to \infty} r_i(a_1) - r_i(a_2) > 0,
$$

which guarantees the rank-preservation. $\qquad \square$

**Proof of Corollary 1.** This is the direct consequence of Theorem 1 and Proposition 1.

### B.5 Strictly Proper Composite Losses

*Strictly proper composite* losses (Menon & Ong, 2016; Reid & Williamson, 2010) are a well-known class of CPE losses.

**Definition 5** (Strictly proper composite losses). A loss function $\ell$ is a strictly proper composite with invertible *link function* $f$ if the minimizer $g^*$ of $R_\ell(g)$ is $f \circ \eta$, where $\circ$ is the composition operator.

If a loss is a strictly proper composite, we can ensure the exact recovery of the class probability by $\eta(a_1, a_2) = f^{-1} \circ g^*(a_1, a_2)$. Also, if we know the relationship between the class probability and the reward, e.g., in the BT, we can recover the exact reward. Some losses, such as the logistic, squared, and exponential, are strictly proper composite.

Table 4: Test reward accuracy. Average over 5 seeds.

| Loss | $\varepsilon = 0.0$ | $\varepsilon = 0.1$ | $\varepsilon = 0.2$ | $\varepsilon = 0.3$ | $\varepsilon = 0.4$ |
|---|---|---|---|---|---|
| logistic | 89.8% (1.0) | 80.8% (1.2) | 73.9% (1.4) | 66.2% (0.7) | 58.3% (0.4) |
| hinge | 89.4% (0.6) | 81.2% (0.4) | 74.5% (0.8) | 66.5% (1.0) | 57.6% (1.6) |
| unhinged | 77.7% (1.4) | 76.9% (1.8) | 74.0% (1.9) | 70.0% (0.9) | 61.6% (2.0) |
| ramp | 92.2% (0.7) | 89.9% (1.0) | 86.8% (1.4) | 82.2% (1.3) | 70.8% (1.4) |
| sigmoid | 91.2% (0.2) | 88.9% (0.4) | 85.6% (1.1) | 80.9% (3.0) | 68.8% (1.6) |

## C  Supplementary Results

### C.1  MNIST Preference Dataset

We present the result of the MNIST preference dataset with standard deviation in Table 4.

**Learning curves.**  We plot the learning curves of the test reward accuracy and the reward margin, defined as $|r(a_1) - r(a_2)|$, on the MNIST Preference dataset for noise levels $\varepsilon \in \{0.0, 0.2, 0.4\}$ in Fig. 4. Across all noise levels, the symmetric losses, especially Ramp and Sigmoid, maintain higher reward accuracy throughout training than the non-symmetric losses. In contrast, the performance of logistic and hinge losses degrades more noticeably as training proceeds under noisy labels. We also observe that the reward margin is consistently larger for the symmetric losses, indicating better separation between preferred and dispreferred samples. This tendency is particularly pronounced for the Sigmoid loss, whose monotone decreasing symmetric form is associated with a larger margin during training.

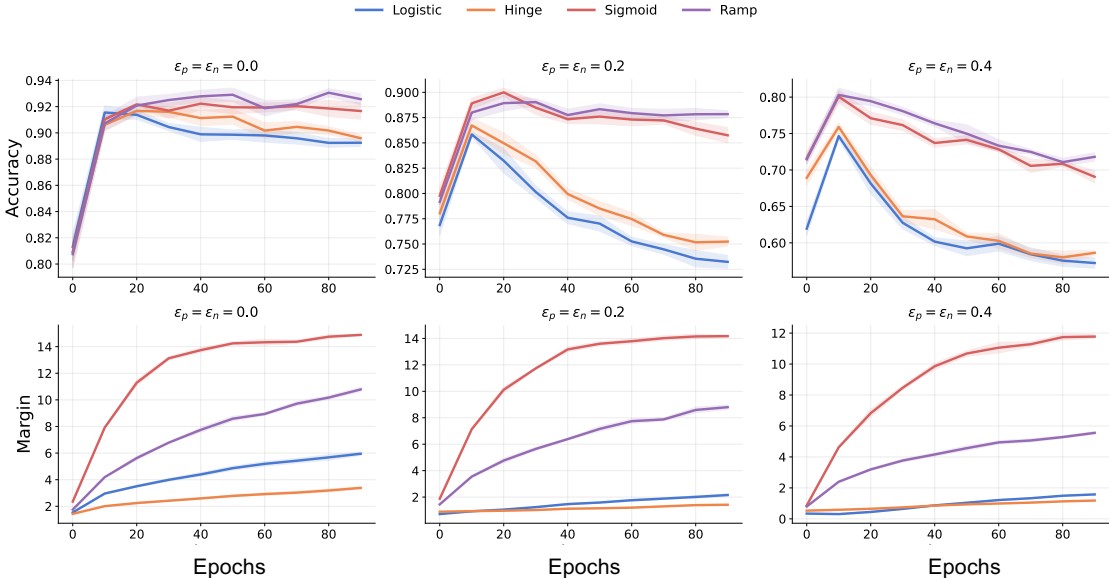

Figure 4:  Learning curves of the test reward accuracy and the reward margin on the MNIST Preference dataset for noise levels $\varepsilon \in \{0.0, 0.2, 0.4\}$.

### C.2  Language Model Alignment

**Sensitivity analysis for the clipping value.** To evaluate the sensitivity of SymPO to the clipping threshold, for sigmoid and ramp losses, we swept the clipping value over $\{10, 20, 30\}$ for noise rates $\varepsilon = 0.2$ and $\varepsilon = 0.4$ on Alpaca Comparison and UFB. We used GPT-4.1-nano and GPT-4o-mini as evaluators. For each setting, we generated responses using 10 random seeds and performed paired t-tests over win rates for all pairs of clipping values. As shown in Table 5, no statistically significant differences were observed in 22 of the 24 settings at the 5% level. This suggests that the clipping threshold is not a sensitive hyperparameter, but instead serves primarily as a practical safeguard against numerical overflow.

| Dataset | Evaluator | Noise | Loss | Clip=10 | Clip=20 | Clip=30 | Sig. pair(s) |
|---------|-----------|-------|------|---------|---------|---------|--------------|
| Alpaca | GPT-4.1-nano | 0.2 | Sigmoid | 65.0 ±2.7 | 64.3 ±2.5 | 64.7 ±2.0 | – |
| | | 0.2 | Ramp | 64.0 ±2.3 | 64.0 ±2.2 | 64.1 ±2.3 | – |
| | | 0.4 | Sigmoid | 63.4 ±2.6 | 63.7 ±2.7 | 63.0 ±3.0 | – |
| | | 0.4 | Ramp | 63.5 ±2.7 | 63.1 ±3.0 | 62.8 ±2.8 | 10 vs 30 |
| | GPT-4o-mini | 0.2 | Sigmoid | 71.3 ±2.3 | 71.0 ±2.3 | 71.5 ±1.8 | – |
| | | 0.2 | Ramp | 69.5 ±2.3 | 69.8 ±1.8 | 69.6 ±2.4 | – |
| | | 0.4 | Sigmoid | 69.0 ±2.9 | 69.4 ±2.2 | 69.1 ±2.9 | – |
| | | 0.4 | Ramp | 67.8 ±2.3 | 67.7 ±2.5 | 67.6 ±2.6 | – |
| UFB | GPT-4.1-nano | 0.2 | Sigmoid | 58.9 ±1.2 | 59.1 ±1.5 | 58.9 ±1.3 | – |
| | | 0.2 | Ramp | 56.4 ±1.1 | 57.1 ±1.7 | 57.1 ±1.0 | – |
| | | 0.4 | Sigmoid | 57.2 ±0.8 | 56.8 ±1.2 | 56.8 ±1.4 | – |
| | | 0.4 | Ramp | 55.9 ±1.2 | 55.9 ±1.3 | 56.0 ±1.4 | – |
| | GPT-4o-mini | 0.2 | Sigmoid | 63.8 ±1.2 | 64.4 ±1.4 | 64.2 ±1.4 | 10 vs 20 |
| | | 0.2 | Ramp | 61.1 ±1.0 | 61.6 ±1.3 | 61.3 ±1.0 | – |
| | | 0.4 | Sigmoid | 61.4 ±1.3 | 61.3 ±1.0 | 61.1 ±1.5 | – |
| | | 0.4 | Ramp | 60.3 ±1.1 | 60.1 ±0.9 | 60.0 ±1.5 | – |

Table 5: Sensitivity of SymPO to the clipping threshold. We swept the clipping value over $\{10, 20, 30\}$ on Alpaca Comparison and UFB, under noise ratios 0.2 and 0.4, using GPT-4.1-nano and GPT-4o-mini as evaluators. Each entry reports win rate (%) ± standard deviation over 10 seeds. The last column shows pairs of clipping values with statistically significant differences under a paired t-test at the 5% level. Only 2 of the 24 settings exhibited any statistically significant difference.

**Sweep over the temperature parameter $\beta$.** We conduct a small hyperparameter sweep over the temperature parameter $\beta$ to examine its effect on SymPO. In many DPO-based implementations, $\beta = 0.1$ is commonly used as the default setting. To verify whether this choice remains appropriate for our symmetric-loss formulation, we evaluate several values of $\beta$. Specifically, we test $\beta \in \{0.1, 0.3, 0.5\}$ on the Alpaca Comparison dataset under two noise levels ($\varepsilon = 0.2$ and $\varepsilon = 0.4$). For each configuration, we generate responses using 10 random seeds and report the mean and standard deviation of the win rate against the SFT model. Judgments are made by GPT-4.1-nano. We set the clipping value to 20. Table 6 reports the results. Overall, $\beta = 0.1$ consistently performs best or nearly best across both noise settings for the symmetric losses. These results support our choice of $\beta = 0.1$ in the main experiments.

**Reward margin.** We plot validation reward-margin curves, $|r(a_{\text{chosen}}) - r(a_{\text{rejected}})|$, on Alpaca Comparison and UFB under noise levels $\varepsilon \in \{0.0, 0.2, 0.4\}$, where the implicit reward is $r(a) = \beta \log \frac{\pi(a)}{\pi_{\text{ref}}(a)}$ (Fig. 5). At higher noise levels, symmetric losses yield larger margins than DPO (logistic loss), indicating stronger separation between chosen and rejected responses. Among them, Sigmoid often achieves the largest margin, suggesting stronger preference contrast for policy optimization. All curves are smoothed with a moving average of 20 evaluation points.

Table 6: Sensitivity analysis for the temperature parameter $\beta$ on the Alpaca Comparison dataset. Performance is measured by the win rate against the SFT model. Judgments are made by GPT-4.1-nano.

| Method | Noise ($\varepsilon$) | $\beta$ | Win rate (%) |
|--------|------------|--------|--------------|
| Sigmoid | 0.2 | 0.1 | 64.3 ±2.5 |
| | | 0.3 | 64.2 ±2.2 |
| | | 0.5 | 64.1 ±2.2 |
| | 0.4 | 0.1 | 63.8 ±2.7 |
| | | 0.3 | 63.0 ±2.7 |
| | | 0.5 | 61.4 ±2.3 |
| Ramp | 0.2 | 0.1 | 64.0 ±2.2 |
| | | 0.3 | 63.0 ±2.4 |
| | | 0.5 | 61.9 ±2.4 |
| | 0.4 | 0.1 | 63.1 ±3.0 |
| | | 0.3 | 59.4 ±3.0 |
| | | 0.5 | 60.2 ±2.9 |

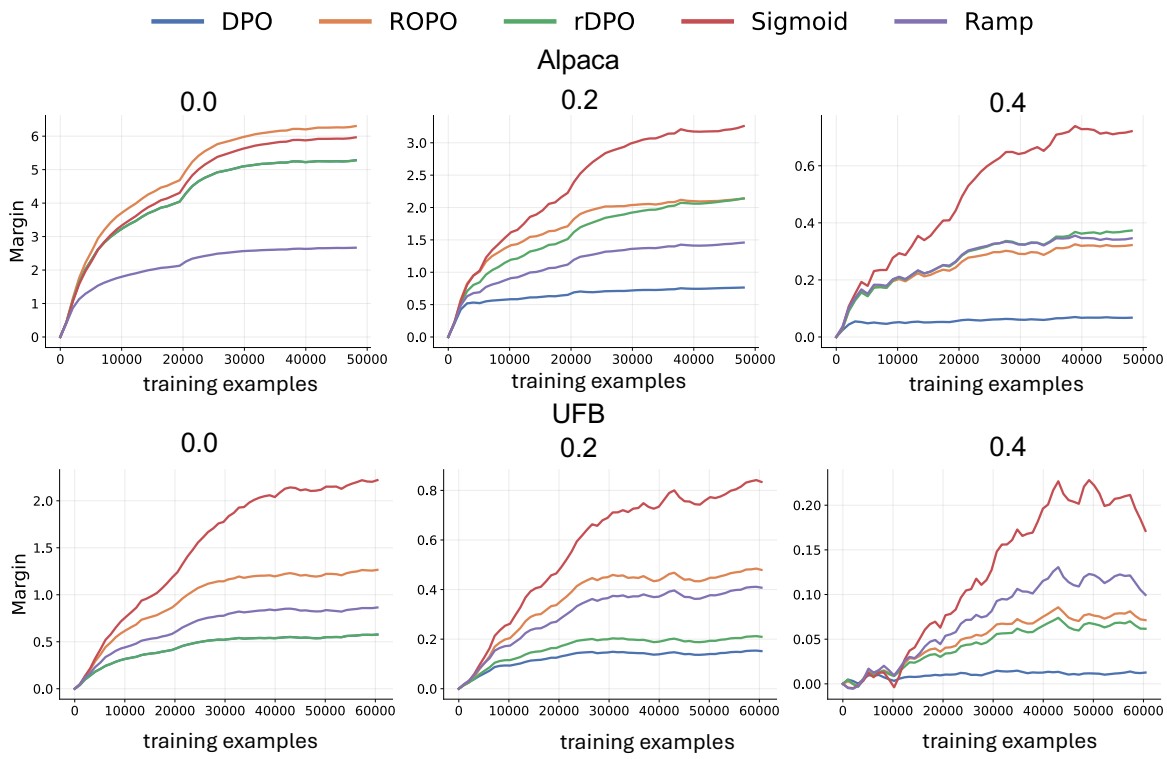

Figure 5: Learning curves of the validation reward margin on Alpaca Comparison and UFB for noise levels $\varepsilon \in \{0.0, 0.2, 0.4\}$. The x-axis is the W&B step (i.e., the number of training examples seen). Validation metrics are logged every 1024 training examples. Curves are smoothed with a moving average over 20 validation points.

