# OpenReview forum: "On Symmetric Losses for Policy Optimization with Noisy Preferences"
_TMLR — Accepted by TMLR_

### Review · Reviewer_j5sf · 2026-02-23

**Summary Of Contributions:**

The paper proposes SymPO, which connects asymmetric noise with symmetric loss functions and proves the learnability of the optimal ranking. It further validates the effectiveness of the algorithm through empirical experiments.

**Audience:**

Yes

**Audience Explanation:**

This topic has received broad attention from the community, as policy optimization plays a crucial role in training and aligning large language models.

**Claims And Evidence:**

Yes

**Claims Explanation:**

The paper provides a rigorous theoretical analysis. However, the experimental section is not entirely convincing. On the Alpaca dataset, the proposed method does not show improvement over existing approaches such as rDPO and ROPO. In addition, the definition of $R_l^\star$ on page 5 appears to require normalization; otherwise, simple rescaling could lead to degeneracy—for example, $\alpha R_l^\star$  could always outperform $R_l^\star$ for some $\alpha$.

**Requested Changes:**

Typo: "approaches (Duchi & Namkoong, 2019; 2021). citebukharin2024robust considered a specific..."

Is there any function satisfying all three properties in Table 1?

Please report the model performance before training for Table 3. This can reflect the improvement of PO.

For problems involving noisy preference data, some prior works, e.g., https://arxiv.org/pdf/2502.14560, adopt data filtering or data selection strategies to mitigate noise. It would be helpful to include a discussion of these approaches. How does your method compare to such filtering-based methods? What specific advantages does your approach offer?

---

> ### Author Response · Authors · 2026-03-06
>
> We thank the reviewer for the positive assessment and helpful suggestions. We revised the paper accordingly, and all modifications are highlighted in blue in the revised manuscript.
>
> **On weaknesses**
>
> > W1: The experimental section is not entirely convincing, especially on Alpaca.
>
> We agree that, on Alpaca, our method does not consistently outperform existing robust methods such as rDPO and ROPO. As clarified in the revised paper, we observed no statistically significant difference in win rates among the robust variants on Alpaca, indicating that our symmetric losses perform comparably to existing robust methods. In contrast, on UFB—especially under higher noise levels—our method achieves the strongest performance among the compared approaches. We believe these results support our main claim that symmetric losses can provide robustness in policy optimization under noisy preference data.
>
> > W2: In addition, the definition of $R_l^\star$ on page 5 appears to require normalization; otherwise, simple rescaling could lead to degeneracy—for example, $\alpha R_l^\star$ could always outperform $R_l^\star$ for some $\alpha$
>
>  We interpret it as pointing out that, for strictly decreasing losses, positively rescaling the reward function $r$ can reduce the absolute value of $R_\ell(r)$ without changing the induced ordering. We agree that this can happen. However, this does not affect our main theoretical claim, because our analyses in Secs. 5.1 and 5.2 only require recovering the correct pairwise ranking, i.e., the sign of $r(a_1)-r(a_2)$, rather than the absolute scale of the reward. Positive rescaling leaves this sign unchanged and thus preserves the decision boundary. If this interpretation differs from the reviewer’s intention, we would appreciate further clarification.
>
>
> **On requested changes**
> > Typo: "approaches (Duchi & Namkoong, 2019; 2021). citebukharin2024robust considered a specific..."
>
> Thank you for catching this. We fixed the typo in the revision.
>
> > Is there any function satisfying all three properties in Table 1?
>
> No. In general, symmetric losses are not class-probability-estimation (CPE) losses, and thus no function satisfies all three properties simultaneously. We now clarify this point in Sec. 5.3 and refer the reader to Theorem 8 of [1].
>
> > Please report the model performance before training for Table 3.
>
> We apologize for the confusion. The win rates reported in Table 3 are measured against the reference (chosen) response. To make the policy improvement clearer, we also report the win rate of the SFT model in the tables.
>
> > For problems involving noisy preference data, some prior works adopt data filtering or data selection strategies to mitigate noise. It would be helpful to include a discussion of these approaches. How does your method compare to such filtering-based methods? What specific advantages does your approach offer?
>
> Our focus in this work is on robust loss design, and thus we did not incorporate data filtering. However, filtering-based methods and robust loss functions are complementary rather than mutually exclusive, as also exemplified by [2]. A practical advantage of our approach is that it improves robustness purely through the loss design, without requiring an additional filtering or selection stage. In this sense, data filtering may further improve practical performance by reducing the effective noise ratio, while our method improves robustness at the loss level. Exploring combinations of diverse symmetric losses and filtering methods is an interesting direction for future work. We added this discussion in Sec. 2.2.
>
> Again, thank you very much for your careful reading and helpful suggestions.
>
> **References**
> [1] Charoenphakdee, Nontawat, Jongyeong Lee, and Masashi Sugiyama. "On symmetric losses for learning from corrupted labels." *International Conference on Machine Learning*. PMLR, 2019.
> [2] Liang, Xize, et al. "ROPO: Robust Preference Optimization for Large Language Models." *arXiv preprint* arXiv:2404.04102, 2024.

---

> > ### Comment · Reviewer_j5sf · 2026-03-06
> >
> > My point is that $R_l^\star$ may degenerate in your definition. Let's assume $r^\star=\arg \inf R_l(r)$. I'm afraid that $R_l(\alpha r^*)$ may smaller than $R_l^\star$ when $\alpha$ is 0 or $\infty$. An easy way to solve my concern is to give an example showing that, in some cases, it won't degenerate.

---

> ### Author Response · Authors · 2026-03-09
>
> Thank you very much for your clarification.
>
> We now understand your point. Indeed, for losses that are unbounded below, such as the unhinged loss $\ell(\alpha) = -\alpha$, the risk can be made arbitrarily small by rescaling, so the optimal risk $R_\ell^\star := \inf_rR_\ell(r)$ may equal $-\infty$.
>
> However, this does not invalidate the definition itself: $R_\ell^\star$ remains well-defined as an infimum in the extended real line, even in such cases. In contrast, most standard classification losses, including the ramp and sigmoid losses that are the main symmetric losses considered in our paper, are lower-bounded. Hence, the corresponding risk is also bounded from below, and therefore $R_\ell^\star > -\infty$.

---

### Review · Reviewer_nEnN · 2026-02-24

**Summary Of Contributions:**

*Summary*

This work proposes SymPO, a new preference alignment framework to deal with potentially mis-labeled preference pairs. They apply a new symmetric losses are used because they are robust to label noise, with both theoretical justification and empirical evidence. Theoretically, the authors argue that rank preservation of the learned reward is sufficient for policy improvement, and show that minimizing risk with classification-calibrated losses induces a rank-preserving reward. Experiments include synthetic MNIST-preference data and LLM alignment on Alpaca Comparison and UltraFeedback Binarized using Llama-3.2-3B-Instruct, with other baseline losses are compared with.

---

*Strength*

- The theory somehow justified the loss construction, which is pretty interesting. Also, reward modeling as binary classification makes the method and theory easy to relate to existing literatures.
- The overall work is well-written and presentation is clear.
- Some empirical evaluations to justify the effectiveness.


---

*Weaknesses*

- The empirical performance seems a bit marginal, with mostly 1pp better than the baseline and some even degraded. Also note that the evaluation are done with some less robust judges.
- The LM experiments primarily vary $\varepsilon$ via random flipping (symmetric corruption), whereas the motivating cases are asymmetric&systematic biases. This reads like a mis-match between theory emphasis and main LLM noise injection. See requested changes.
- Some baseline choices such as ROPO are simplified, i.e., ROPO by neglecting the rej-sampling part by only using its loss function.
- Some theoretical results seem intuitive.

**Audience:**

Yes

**Audience Explanation:**

Yes.

Dealing mis-labeled noisy pairs is an important challenge for the alignment community, which has been studied by wide-range of works. This work proposes an interesting idea through reward function symmetry and provide specific theory that links the formulation to the practice, which I believe many readers will be benefited from reading it.

**Broader Impact Concerns:**

Not applied here.

**Claims And Evidence:**

Yes

**Claims Explanation:**

Yes.

Most claims are explained theoretically, along with empirical experiment to justify its actual effectiveness.

**Requested Changes:**

Some issue regarding presentation, clarity, and minor change that can be easily addressed:

- I think the definition of noisy pairs in alignment is not clearly distinguished from the other works. Many works have defined noisy pairs as the one with small likelihood margin (e.g., [1] [openreview.net/forum?id=0lNwIIHWhZ](https://openreview.net/forum?id=0lNwIIHWhZ), NeurIPS 2025; [2] [openreview.net/forum?id=uaMSBJDnRv](https://openreview.net/forum?id=uaMSBJDnRv), ICLR 2025). To improve the clarity, section 2.2 should add the discussion of noisy pairs in these works.
- Could you add a multiplication notation between $y$ and function $g$ in the Eq above Eq.6? The presentation looks a bit weird.
- What is the base model that WR compare against (is it the SFT model or raw model)? I may miss something but I didn't find it in the main text. Please add this detail to the caption of Table 3.

Some further issues:
- Following from weakness 2, please consider how to address that mis-match.
- Since the loss is new proposed, and some previous analysis over DPO may no longer be applied to here. I suggest add some theoretical analysis over $\beta$ or empirical results to better help reader understand the hyperparameters. I think $\beta=0.1$ in DPO and $\beta=0.1$ in your method have different effect.
- As the context is mostly for offline alignment, I was wondering any potential direction that can be extended to online RLHF reward design. If there is, please add to section 7.
- Since most of the results are quantitative, I was wondering if any qualitative training metric for in-training curve that can be provided to better understand the learning dynamics.

---

> ### Author Response · Authors · 2026-03-09
>
> We thank the reviewer for the constructive feedback. We have revised the paper accordingly, and the modified parts are highlighted in blue.
>
> **On the weaknesses.**
>
> > W1: The empirical performance seems a bit marginal, with mostly 1pp better than the baseline and some even degraded. Also note that the evaluation are done with some less robust judges.
>
> We agree that the empirical gains are modest. Our goal is not to claim universal dominance in clean settings, but to demonstrate **robustness under noisy preferences**. From this perspective, SymPO is competitive across datasets, and on UFB the sigmoid variant achieves the strongest results.
>
> > W2: The LM experiments primarily vary via random flipping (symmetric corruption), whereas the motivating cases are asymmetric and systematic biases. This reads like a mismatch between theory emphasis and main LLM noise injection.
>
> We agree that this point needs clearer explanation. As shown in **Lemma 1**, in preference learning, asymmetric noise can be reduced to an equivalent symmetric noise model. Therefore, symmetric random flipping provides a controlled and theoretically justified setting for validating the robustness predicted by our analysis. We have added this explanation to the experiment section.
>
> > W3: Some baseline choices such as ROPO are simplified, i.e., ROPO by neglecting the rejection-sampling part and only using its loss function.
>
> This is intentional: our comparison is designed at the loss-function level. Since SymPO is primarily a loss-design contribution, we implemented ROPO’s loss within the same training pipeline to enable a controlled comparison, rather than introducing additional factors such as rejection sampling. We also added a discussion in Sec. 2.2 on the possibility of combining our approach with other methods.
>
> > Some theoretical results seem intuitive.
>
> While notions such as rank preservation and classification calibration are individually known, we believe the main contribution is to connect them to policy improvement in RLHF / offline preference optimization. This provides a principled bridge from binary classification theory to (robust) preference optimization.

---

> > ### Author Response · Authors · 2026-03-09
> >
> > **On the requested changes.**
> >
> > > I think the definition of noisy pairs in alignment is not clearly distinguished from the other works. Many works have defined noisy pairs as the one with small likelihood margin (e.g., [1], [2] ).
> >
> > We agree. Another line of work considers **instance-dependent noise**, where errors arise from ambiguous pairs with small likelihood margins. In contrast, our work focuses on **instance-independent noise**, which models systematic mislabeling in preference data. To improve clarity, we extended Section 2.2 to distinguish these two settings and discussed the reviewer-cited works under the instance-dependent noise category.
> >
> > > Could you add a multiplication notation between $y$ and function $g$ in the Eq above Eq.6? The presentation looks a bit weird.
> >
> > Thank you for catching this. We corrected the notation.
> >
> > > What is the base model that WR compare against (is it the SFT model or raw model)? I may miss something but I didn't find it in the main text. Please add this detail to the caption of Table 3.
> >
> > Apologies for the confusion. The comparison in Table 3 is against the reference response (chosen response). We now clarify this explicitly in the main text and in the caption of Table 3. We also added the win rate of the SFT model to make the policy improvement clearer.
> >
> > > Since the loss is new proposed, and some previous analysis over DPO may no longer be applied to here. I suggest add some theoretical analysis over $\beta$ or empirical results to better help reader understand the hyperparameters.
> >
> > We followed the common choice $\beta=0.1$, which is widely used in DPO-style implementations [1]. To further justify this choice, we added a brief sensitivity analysis over several $\beta$ values in Table 8. In our experiments, $\beta=0.1$ performed best or near the best among the tested values.
> >
> > > As the context is mostly for offline alignment, I was wondering any potential direction that can be extended to online RLHF reward design. If there is, please add to section 7.
> >
> > We agree that extending symmetric losses to **online RLHF** is an important direction. However, due to the substantial computational cost, conducting such experiments during the rebuttal period is difficult. Since our goal here is to verify robust policy optimization under noisy preferences, we believe offline preference optimization provides a sufficient and controlled setting for this purpose. We have added online RLHF as a future direction in the conclusion.
> >
> > > Since most of the results are quantitative, I was wondering if any qualitative training metric for in-training curve that can be provided to better understand the learning dynamics.
> >
> > To better illustrate the learning dynamics, we added plots of the reward accuracy and the margin for the MNIST preference dataset. We find that symmetric losses, especially the sigmoid loss, tend to produce larger margins than non-symmetric losses. For language model alignment, we also report the margin $r(a_{\text{chosen}})-r(a_{\text{rejected}})$ and observe a consistent trend for symmetric losses.
> >
> > Again, thank you very much for your careful reading and helpful suggestions.
> >
> > **Reference** [1] https://github.com/huggingface/trl/blob/main/trl/trainer/dpo_config.py

---

> > > ### Comment · Reviewer_nEnN · 2026-03-10
> > >
> > > Many thanks for the response.
> > >
> > > Regarding weakness 1, I feel bring Table 5 and 6 to the main text could make the evidence better, as even the improvement is not notable but it's statistical significant.
> > >
> > > As said in my previous review, I think the method is simple and interesting, which many readers will benefit from reading it. The additional empirical results and explanation also improves the manuscript. Therefore, I am happy to recommend this paper for acceptance.

---

> ### Author Response · Authors · 2026-03-11
>
> Thank you very much for your positive comments and helpful suggestion. We agree that placing Tables 5 and 6 in the main text would make the empirical evidence clearer. To keep the rebuttal consistent with our responses to the other reviewers, we will reflect this change in the post-rebuttal revision.

---

### Review · Reviewer_dieo · 2026-02-26

**Summary Of Contributions:**

## Summary

This work introduces **Symmetric Preference Optimization (SymPO)**, a robust policy optimization method designed to handle asymmetric noisy preference data. The authors formally prove that, in the context of reward modeling, asymmetric preference noise is theoretically equivalent to symmetric noise. This insight allows them to employ **symmetric losses** from binary classification for offline preference optimization. The effectiveness of SymPO is validated through both synthetic MNIST experiments and real-world language model alignment tasks.


---

## Strengths

**Theoretical Foundation**: The work provides rigorous theoretical proofs establishing the equivalence between asymmetric and symmetric preference noise (Lemma 1), and clarifies why rank-preserving rewards are sufficient to guarantee policy improvement.

**Ease of Implementation**: The loss functions (Ramp and Sigmoid) are simple and can be readily integrated into existing preference optimization pipelines.


---

## Weaknesses


**Conceptual and Theoretical Limitations**
The core idea—that ranking is what matters for RLHF—is a common intuition in the field. Much of the mathematical framework is adapted from established classification theory rather than being entirely novel.

**Practical and Numerical "Cracks"**
Transitioning to practice reveals stability issues; unlike the naturally stable logistic loss in standard DPO, the authors had to manually clip logit values to [-20, 20] for SymPO. Additionally, some symmetric losses are non-convex, which can complicate optimization and lead to local minima.

**Audience:**

Yes

**Audience Explanation:**

The findings of this paper would be of interest to the TMLR audience, which consists of researchers and practitioners in machine learning, particularly those focused on large language model alignment

**Broader Impact Concerns:**

The submission includes a brief discussion on social impact.

**Claims And Evidence:**

Yes

**Claims Explanation:**

The claims in the submission are supported by a combination of rigorous theoretical proofs and diverse empirical evidence. The authors provide a clear bridge between established binary classification theory and the practical needs of language model alignment.

**Requested Changes:**

In Section 6.2, the logit values were clipped to [-20, 20] for the Ramp and Sigmoid losses to ensure numerical stability. Given that standard DPO is inherently more stable due to its convex logistic form, can you provide a sensitivity analysis or ablation study on this clipping range. If performance is highly sensitive to this value, it potentially undermines the claim that SymPO avoids "additional hyperparameters to be tuned" compared to existing methods.

---

> ### Author Response · Authors · 2026-03-07
>
> We thank the reviewer for the constructive comments. We have revised the paper accordingly, and the corresponding modifications are highlighted in blue.
>
> **On weaknesses**
> > W1:The core idea—that ranking is what matters for RLHF—is a common intuition in the field. Much of the mathematical framework is adapted from established classification theory rather than being entirely novel.
>
> We agree that the importance of ranking and the role of classification-calibrated losses are well known in RLHF and binary classification respectively. Our contribution is not to claim either of these ideas as new in isolation, but to formalize their connection in the RLHF / preference-optimization setting. In particular, we show that classification-calibrated losses induce rank-preserving rewards, and that rank preservation is sufficient for policy improvement. We believe that making this connection explicit helps clarify why a broad class of losses from binary classification is theoretically justified for preference optimization.
>
> > W2: Additionally, some symmetric losses are non-convex, which can complicate optimization and lead to local minima.
>
> We agree that Ramp and Sigmoid are non-convex and may be less favorable than the logistic loss from an optimization perspective. However, their main advantage is robustness to noisy preferences, which is precisely the regime we target. Empirically, this trade-off is favorable in our experiments, especially under higher noise levels, where SymPO is competitive with or outperforms existing robust baselines.
>
> **On requested change**
> > On clipping and numerical stability (raised both in the weaknesses and in the requested changes):
>
> We agree that this point deserves clarification. To address it, we conducted an additional sensitivity analysis by sweeping the clipping range over {10, 20, 30} for both sigmoid and ramp losses, across Alpaca Comparison and UFB, noise rates 0.2 and 0.4, and both evaluators (GPT-4.1-nano and GPT-4o-mini), yielding 24 settings in total. Using a 5% t-test, we found statistically significant differences in only 2 of the 24 settings; the remaining 22 settings showed no significant difference (reported in Table 7). This suggests that the clipping range does not behave as a delicately tuned hyperparameter, but rather as a practical device to prevent numerical overflow. We have added this ablation to the revision.
>
> We also clarified an implementation detail that was not accurately described in the original manuscript. Although the paper stated that the clipping value was 20.0 for all settings, the actual value used for UFB was 10, while Alpaca used 20. Since the sensitivity analysis indicates that the clipping range has negligible influence on the results, we replaced the main UFB results in Tables 3, 5, and 6 with runs using a clipping value of 20.0 and Section 6.2 now clearly states that a clipping value of 20.0 is used for all settings.
>
> Again, thank you very much for your careful reading and helpful suggestions.

---

### Decision · Action_Editor_3KXM · 2026-03-23

**Recommendation:** Accept with minor revision

**Additional Comments:**

- One reviewer raised a point about the reward definition not being meaningful everywhere. The authors provided a rebuttal is noted in a footnote on Page 8. The authors are encouraged to include this technical limitation in their statement for Theorem 2 to help the reader get a complete picture

- The experimental results table with standard deviation appears to be in the appendix. The authors should consider moving this table to the main paper. Furthermore, the use of bold to highlight best performing models should be consistent. Current Table 3 shows rDPO also have the same value as Sigmoid but only SymPO's method is in bold.

- Please integrate all text provided during rebuttal in the paper.

**Audience:**

Yes

**Audience Explanation:**

The method and the analysis in the paper is on the topic of preference optimization which is an area of interest to many members in the TMLR community especially those working on language model alignment.

**Claims And Evidence:**

Yes

**Claims Explanation:**

The paper proposes an offline preference optimization method called SymPO (Symmetric Preference Optimization) that is robust to noise in preference datasets . The paper observes that the noise in preference dataset while being asymmetric can be treated as symmetric noise by making use of the fact that scoring functions in preference optimization have anti-symmetry. The observation noted above allows for the use of symmetric loss objectives in binary classification that are known to be robust to label noise. The paper establishes that the reward resulting from using symmetric loss objectives is rank preserving that is sufficient for policy improvement. Empirical evidence with synthetic data (MNIST) data is provided to support robustness to noise in preference data while experiments with a LLM (Llama-3.2 3B parameters) is provided to support the performance of proposed method on language model alignment.

The claims made in the paper are supported by mathematical arguments that are correct (Lemma 1, Proposition 1, Theorems 1 & 2). The main claims are:
- Equivalence between asymmetric and symmetric noise via the asymmetry in scoring preference pairs.
- Symmetric losses preserve risk ordering under symmetric noise. This leads to SymPO
- That minimizing symmetric loss is results in a reward that's rank preserving that is sufficient for policy improvement.

Additionally, the experimental evidence is instructive as it shows robustness to noise as well as performance on a language model alignment task. A major strength of this paper is that the developed method has sufficient motivation to be interesting to many members in the TMLR community

All three reviewers agree that the paper provides accurate, convincing and clear evidence for the analysis and that the proposed method is simple and easy to implement. The recommendations are:
- 1 Accept
- 2 Leaning Accepts

Given the weight of evidence and reviewers support, the AE recommends an accept (with minor revision) as well.